



# A seasonal algorithm of the snow-covered area fraction for mountainous terrain

Nora Helbig [1], Michael Schirmer [1], Jan Magnusson [2], Flavia Mäder [1,3], Alec van Herwijnen [1], Louis Quéno [1], Yves Bühler [1], Jeff S. Deems [4], and Simon Gascoin [5]

[1]WSL Institute for Snow and Avalanche Research SLF, Davos, Switzerland
[2]Statkraft AS, Oslo, Norway
[3]Institute of Geography, University of Bern, Bern, Switzerland
[4]National Snow and Ice Data Center, University of Colorado, Boulder, CO, USA
[5]Centre d'Etudes Spatiales de la Biosphère, CESBIO, Univ. Toulouse, CNES/CNRS/INRAE/IRD/UPS, 31401 Toulouse, France

**Correspondence:** Nora Helbig (norahelbig@gmail.com)

**Abstract.** The snow cover spatial variability in mountainous terrain changes considerably over the course of a snow season. In this context, fractional snow-covered area ($fSCA$) is therefore an essential model parameter characterizing how much of the ground surface in a grid cell is currently covered by snow. We present a seasonal $fSCA$ algorithm using a recent scale-independent $fSCA$ parameterization. For the seasonal implementation we track snow depth ($HS$) and snow water equiva-

lent ($SWE$) and account for several alternating accumulation-ablation phases. Besides tracking $HS$ and $SWE$, the seasonal $fSCA$ algorithm only requires computing subgrid terrain parameters from a fine-scale summer digital elevation model. We implemented the new algorithm in a multilayer energy balance snow cover model. For a spatiotemporal evaluation of modelled $fSCA$ we compiled three independent $fSCA$ data sets. Evaluating modelled 1 km $fSCA$ seasonally with $fSCA$ derived from airborne-acquired fine-scale $HS$ data, satellite- as well as terrestrial camera-derived $fSCA$ showed overall normalized

root mean square errors of respectively 9 %, 20 % and 22 %, and represented seasonal trends well. The overall good model performance suggests that the seasonal $fSCA$ algorithm can be applied in other geographic regions by any snow model application.

## 1 Introduction

In mountainous terrain, the large spatial variability of the snow cover is driven by the interaction of meteorological variables

with the underlying topography. Over the course of a winter season the dominating topographic interactions with wind, precipitation and radiation vary considerably, which generate the characteristic seasonal dynamics of spatial snow depth variability (e.g. Luce et al., 1999). This spatial variability or how much of the ground is actually covered by snow is typically characterized by the fractional snow-covered area ($fSCA$). $fSCA$ is a crucial parameter in model applications such as weather forecasts (e.g. Douville et al., 1995; Doms et al., 2011), hydrological modelling (e.g. Luce et al., 1999; Thirel et al., 2013; Magnusson

et al., 2014; Griessinger et al., 2016, 2019) or avalanche forecasting (Bellaire and Jamieson, 2013; Horton and Jamieson, 2016; Vionnet et al., 2014) and is also used for climate scenarios (e.g. Roesch et al., 2001; Mudryk et al., 2020).





$fSCA$ can be retrieved from various satellite sensor images such as from Moderate Resolution Imaging Spectroradiometer (MODIS) or Sentinel-2 (e.g. Hall et al., 1995; Painter et al., 2009; Drusch et al., 2012; Masson et al., 2018; Gascoin et al., 2019). However, a temporal and spatial inconsistent coverage due to time gaps between satellite revisits, data delivery and the

frequent presence of clouds requires additional solutions (Parajka and Blöschl, 2006; Gascoin et al., 2015). Though fine-scale spatial snow cover models provide spatial snow depth distributions which could be used to derive coarse-scale $fSCA$, applying such models to larger regions is generally not feasible which is in part due to computational cost, a lack of detailed input data and limitations in model structure or parameters. While some of these limitations can be overcome using current snow cover model advances applying data assimilation routines (e.g. Andreadis and Lettenmaier, 2006; Nagler et al., 2008; Thirel et al.,

2013; Griessinger et al., 2016; Huang et al., 2017; Baba et al., 2018; Griessinger et al., 2019), the inherent uncertainties in input or assimilation data still remain. Computationally efficient subgrid $fSCA$ parameterizations accounting for unresolved snow depth variability, are therefore currently still the method of choice for coarse-scale model systems, such as weather forecast, land surface and earth system models. Furthermore, $fSCA$ parameterizations are essential when assimilating satellite snow-covered area data in model systems (e.g. Zaitchik and Rodell, 2009)

Several compact, closed-form $fSCA$ parameterizations were suggested for coarse-scale model applications (e.g. Douville et al., 1995; Roesch et al., 2001; Yang et al., 1997; Niu and Yang, 2007; Su et al., 2008; Zaitchik and Rodell, 2009; Swenson and Lawrence, 2012). Most of these $fSCA$ parameterizations were heuristically developed. Some parameterizations introduced subgrid terrain parameters (e.g. Douville et al., 1995; Roesch et al., 2001; Swenson and Lawrence, 2012). The $tanh$-form, suggested by Yang et al. (1997), was later confirmed by integrating theoretical log-normal snow distributions and fitting the

resulting parametric depletion curves using the spatial snow depth distribution ($\sigma_{HS}$) in the denominator of fitted $fSCA$ curves (Essery and Pomeroy, 2004). Through advances in remote sensing techniques, fine-scale spatial $HS$ data became more readily available allowing to empirically parameterize $\sigma_{HS}$ in complex topography at peak of winter (PoW) or during accumulation (Helbig et al., 2015b; Skaugen and Melvold, 2019). By parameterizing $\sigma_{HS}$ using subgrid terrain parameters, Helbig et al. (2015b) enhanced the $tanh$-$fSCA$ parameterization of Essery and Pomeroy (2004) by accounting for topographic influence.

Furthermore, Helbig et al. (2020) re-evaluated this empirically derived $fSCA$ parameterization with high-resolution spatially distributed snow depth data sets from 7 different geographic regions at PoW. They introduced a scale-dependency in the dominant scaling variables that improved the empirical $fSCA$ parameterization by making it applicable across spatial scales $\geq 200$ m.

Many studies highlighted that the same mean $HS$ in early winter or in late spring can lead to substantially different $fSCA$

(Luce et al., 1999; Niu and Yang, 2007; Magand et al., 2014), a phenomenon that has led to the introduction of hysteresis in some $fSCA$ parameterizations (e.g. Luce et al., 1999). Previously found interannual time-persistent correlations between topographic parameters and snow depth distributions (e.g. Schirmer et al., 2011; Schirmer and Lehning, 2011; Revuelto et al., 2014; López-Moreno et al., 2017) suggest indeed that a time-dependent $fSCA$ implementation might be feasible. However, a seasonal model implementation of a closed form $fSCA$ parameterization also needs to account for alternating snow ac-

cumulation and melt events during the season. Especially at lower elevations, the separation of the depletion curve in only one accumulation period followed by a melting period is no longer applicable (e.g. Egli and Jonas, 2009). A seasonal $fSCA$



implementation in mountainous regions that accounts for these alternating periods is challenging. While some seasonal $fSCA$ implementations of varying complexities were previously proposed (e.g. Niu and Yang, 2007; Su et al., 2008; Egli and Jonas, 2009; Swenson and Lawrence, 2012; Nitta et al., 2014; Magnusson et al., 2014; Riboust et al., 2019) a detailed evaluation

of seasonally parameterized $fSCA$ with $fSCA$ derived from high-resolution spatial as well as temporal $HS$ data or snow products is currently still missing.

This article presents a seasonal $fSCA$ implementation and its temporal evaluation with high-resolution observation data in various geographic regions throughout Switzerland. The algorithm is based on the $fSCA$ parameterization for complex topography proposed by Helbig et al. (2015b, 2020) and applies two different empirical parameterizations for the spatial snow

depth distribution, namely the ones from Egli and Jonas (2009) and Helbig et al. (2020). The seasonal $fSCA$ algorithm allows for alternating snow accumulation and melt events during the season by accounting for the history of previous $HS$ and $SWE$ values. We implemented the algorithm in a multilayer energy balance snow cover model (modified JIM, the JULES investigation model by Essery et al. (2013)) which we ran with COSMO-1 (operated by MeteoSwiss) reanalysis data, measured $HS$ and RhiresD precipitation data (MeteoSwiss). The seasonal performance of this algorithm was evaluated using daily

modelled 1 km $fSCA$ in Switzerland. For the evaluation we compiled $fSCA$ data sets from terrestrial cameras, airborne surveys and satellite imagery. With this we were able to evaluate modelled $fSCA$ using independent $HS$ data sets in high spatial resolution and snow products in high temporal resolution.

## 2 Fractional snow-covered area algorithm

The $fSCA$ algorithm consists of four parts (cf. upper large box in Figure 1). The first part describes the closed form $fSCA$

parameterization using snow depth $HS$ and standard deviation of subgrid snow depth $\sigma_{HS}$ of a grid cell. The second and third part describe two different $\sigma_{HS}$ parameterizations, one derived for mountainous terrain developed on PoW data ($\sigma_{HS}^{\text{topo}}$) and one for flat terrain developed on accumulation data ($\sigma_{HS}^{\text{flat}}$). These are the inputs to the $fSCA$ function in part one. The fourth part handles the distinctly different paths between $\sigma_{HS}$ and $HS$ during accumulation and ablation periods, the hysteresis. This last part thus describes the technical aspects for a seasonal implementation of $fSCA$, presented in part one, which requires

tracking $HS$ and $SWE$ over the season, deriving extreme values of $HS$ and $SWE$ as well as the two $\sigma_{HS}$ parameterizations presented in part two and three.

### 2.1 $fSCA$ parameterization

We use the $fSCA$ parameterization of Helbig et al. (2015b) derived by integrating a theoretical normal snow depth distribution at PoW, assuming spatially homogeneous melt and by fitting the resulting depletion curves over a range of coefficients of

variation $CV$ (standard deviation divided by its mean) in snow depth ranging from 0.06 to 1.00:

$$fSCA = \tanh(1.3 \frac{HS}{\sigma_{HS}}) \,. \tag{1}$$





Using $\sigma_{HS}$ in Eq. (1) allowed Helbig et al. (2015b) to introduce the close link between spatial snow depth variability and topography in $fSCA$.

Eq. (1) uses current $HS$ in the numerator and $\sigma_{HS}$ at seasonal maximum $HS$ in the denominator, which we adapt here for a
seasonal $fSCA$ algorithm as described in Section 2.4. For the seasonal $fSCA$ algorithm we further compute $\sigma_{HS}$ differently over flat and steep terrain ($\sigma_{HS}^{\text{flat}}$, $\sigma_{HS}^{\text{topo}}$) which is described in the following.

## 2.2  $\sigma_{HS}$ parameterization for mountainous terrain at peak of winter ($\sigma_{HS}^{\text{topo}}$)

Helbig et al. (2020) could use the same functional form to empirically describe the spatial snow depth variability $\sigma_{HS}$ at PoW in mountainous terrain than Helbig et al. (2015b) when using snow data sets from seven different geographic regions and two
continents:

$$\sigma_{HS}^{\text{topo}} = HS^c \mu^d \exp[-(\xi/L)^2] \tag{2}$$

albeit that they introduced scale-dependent parameters $c(L)$ and $d(L)$ in Eq. (2), which enhanced the $\sigma_{HS}$ parameterization across spatial scales for domain sizes $L \geq 200$ m. $\sigma_{HS}^{\text{topo}}$ (Eq. (2)) was parameterized using spatial mean snow depth and subgrid summer terrain parameters: a squared slope related parameter $\mu$ and a terrain correlation length $\xi$ for each domain size $L$
(coarse grid cell). Given that the $\sigma_{HS}$ parameterization in Eq. (2) accounts for the impact of topography on $\sigma_{HS}$, we indicate that with 'topo' ($\sigma_{HS}^{\text{topo}}$). For more details on Eq. (2) we refer to Helbig et al. (2015b, 2020) to keep the focus of this study on the seasonal $fSCA$ algorithm and its evaluation.

## 2.3  $\sigma_{HS}$ parameterization for flat terrain during accumulation ($\sigma_{HS}^{\text{flat}}$)

$\sigma_{HS}^{\text{topo}}$ was developed for grid cells in mountainous terrain. Here, we present a $\sigma_{HS}$ that can be applied in flat terrain, which we
indicate with 'flat' ($\sigma_{HS}^{\text{flat}}$). Egli and Jonas (2009) derived an empirical parameterization for $\sigma_{HS}$ during accumulation by fitting mean and standard deviation of 77 flat field $HS$ measurements distributed throughout Switzerland over six consecutive winter seasons. The resulting parameterization solely uses $HS$ and a constant fit parameter:

$$\sigma_{HS}^{\text{flat}} = HS^{0.839} \ . \tag{3}$$

## 2.4  Seasonal $fSCA$ implementation

For the implementation of our seasonal $fSCA$ algorithm (cf. Eq. 1-3) in any snow cover model, tracking snow information (i.e. keeping the history) through several alternating accumulation-ablation phases is required. By tracking snow information we can use current to extreme $HS$ values to derive $\sigma_{HS}$ (Eq. (2) and (3)) and $fSCA$ (Eq. (1)). We search extreme points in time using $SWE$ to avoid influences of snow settling. Since our $fSCA$ algorithm needs $HS$ as input, the corresponding $HS$
values of $SWE$ extreme points are applied. In the following we will not specify this anymore but instead only refer to extreme



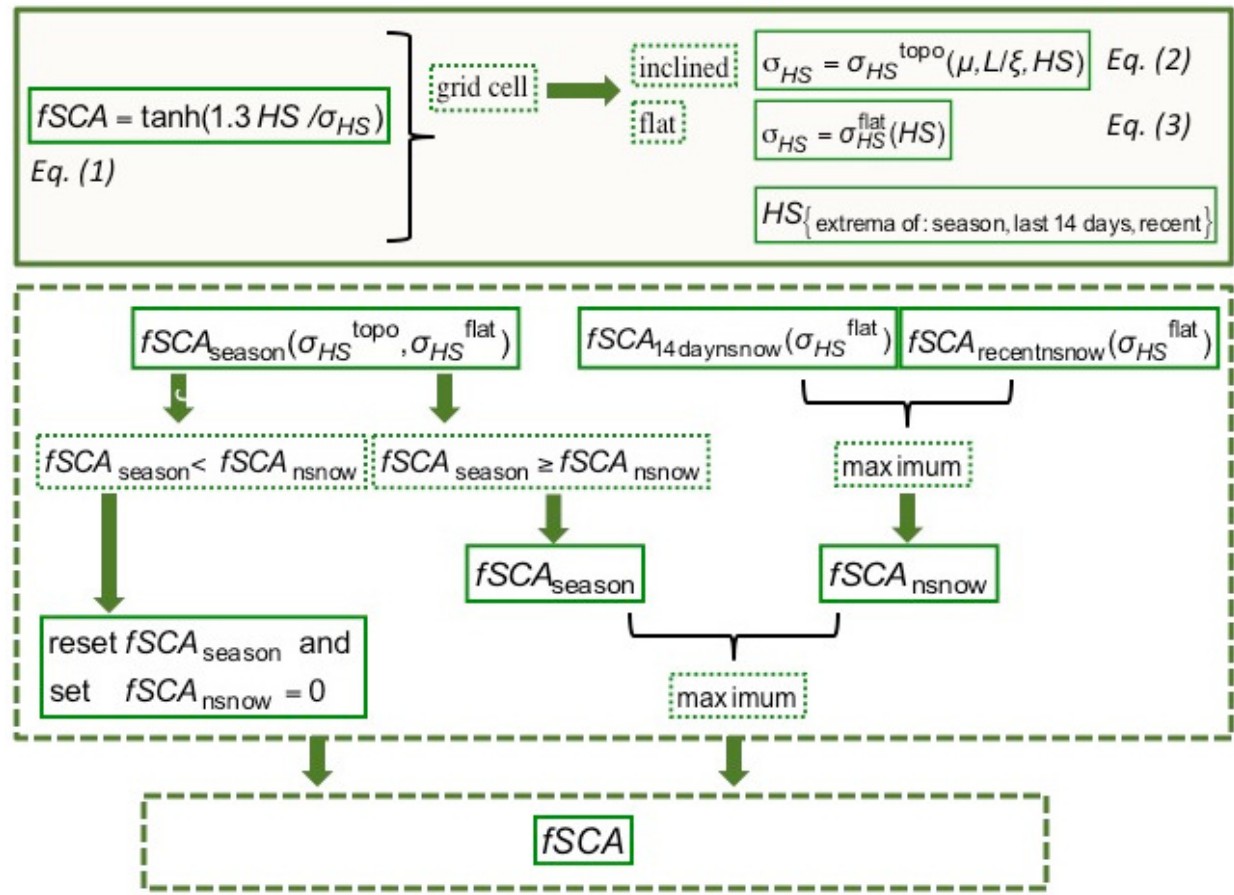

**Figure 1.** Sketch of the seasonal $fSCA$ algorithm as used for one grid cell.

values of $HS$ (minimum, maximum) or $HS$ differences. A full seasonal $fSCA$ algorithm, i.e. including the tracking of $HS$ and $SWE$ over the course of the season, is applied per grid cell of a distributed snow cover model.

Over the course of the season we describe the $fSCA$ curve by means of one seasonal $fSCA$ ($fSCA_\mathrm{season}$) and one $fSCA$ for snowfall events ($fSCA_\mathrm{nsnow}$). This is done to ensure that a snowfall may add significantly to $fSCA$ (i.e. $fSCA_\mathrm{nsnow} >$

$fSCA_\mathrm{season}$) but, once the new snow has started to melt, $fSCA$ can return to similar $fSCA$ values than before. For computing the different $fSCA$ we use Eq. (1) but different $HS$ values (from current to extremes) as well as $\sigma_{HS}$, i.e. $\sigma_{HS}^\mathrm{topo}$ (Eq. (2)) or $\sigma_{HS}^\mathrm{flat}$ (Eq. (3)) (cf. box in the middle in Figure 1). The complete technical aspects of the derivation of all $fSCA$ including some pseudocode are given in Appendix 1.

The final $fSCA$ is obtained from taking the maximum of $fSCA_\mathrm{nsnow}$ and $fSCA_\mathrm{season}$.





## 3  Data

### 3.1  Modelled $fSCA$ and $HS$ maps

We model the snow cover evolution using the JULES investigation model (JIM). JIM is a multi-model framework of physically based energy-balance models solving the mass and energy balance for a maximum of three snow layers (Essery, 2013). While the multi-model framework JIM offers 1701 combinations of various process parameterizations, Magnusson et al. (2015) selected a specific combination that performed best for snow melt modelling for Switzerland, predicting daily snow mass and snowpack runoff for the operational snow hydrology service (OSHD) at WSL Institute of Snow and Avalanche Research SLF. We ran JIM$_{\text{OSHD}}$ in 1 km resolution with hourly meteorological data from the COSMO-1 model (operated by MeteoSwiss) for Switzerland. We used a reanalysis product of daily observed precipitation (RhiresD) from MeteoSwiss as well as COSMO-1 data. Daily $HS$ measurements from manual observers as well as from a dense network of automatic weather stations (AWS) were used to correct precipitation data via optimal interpolation (OI) (Magnusson et al., 2014), which is a computational efficient data assimilation approach. Using OI in JIM$_{\text{OSHD}}$, Griessinger et al. (2019) obtained improved discharge simulations in 25 catchments over four hydrological years.

To describe the subgrid snow cover evolution in mountainous terrain, the seasonal $fSCA$ algorithm was implemented in JIM$_{\text{OSHD}}$. As daily values we use model output generated at 6 am (UTC). In the following, when we refer to modelled $fSCA$ and $HS$ maps we mean $fSCA$ and $HS$ from JIM$_{\text{OSHD}}$ model output.

We additionally computed the snow cover evolution with JIM$_{\text{OSHD}}$ using two simplifications in the seasonal $fSCA$ algorithm (Figure 1). Both simplifications are used in coarse-scale model applications and allow us here to estimate the relevance of applying the full seasonal $fSCA$ algorithm. First, we switched off all new snow $fSCA$ updates, i.e. the final $fSCA$ was set to $fSCA_{\text{season}}$. Second, we defined a $fSCA_{\text{curr}}$ which only uses current modelled $HS$ in $fSCA$ equation (Eq. (1)), i.e. which does not require any $HS$ tracking. We indicate these snow cover simulations with JIM$_{\text{OSHD}}^{\text{season}}$ and JIM$_{\text{OSHD}}^{\text{curr}}$.

### 3.2  Evaluation data

#### 3.2.1  ADS fine-scale $HS$ maps

We used fine-scale spatial $HS$ maps gathered by airborne digital scanning (ADS) with an opto-electronic line scanner on an airplane. Data were acquired over the Wannengrat and Dischma area near Davos in the eastern Swiss Alps (Bühler et al., 2015). We used ADS-derived $HS$ maps at three points in time during one winter season, namely during accumulation at 26 January ('acc'), at approximate peak of winter at 9 March ('PoW') and during ablation season at 20 April 2016 ('abl') (Marty et al., 2019). We used a summer DEM from ADS to derive the snow-free terrain parameters.

Each ADS data set covers about 150 km$^2$ with 2 m spatial resolution. Compared to terrestrial laser scan (TLS)-derived $HS$ data of a subset, the 2 m ADS-derived $HS$ maps had a root mean square error (RMSE) of 33 cm and a normalized median absolute deviation (NMAD) of the residuals (Höhle and Höhle, 2009) of 24 cm (Bühler et al., 2015).



### 3.2.2 ALS fine-scale $HS$ maps

We used fine-scale spatial $HS$ maps gathered by airborne laser scanning (ALS). The ALS campaign was a Swiss partner mission of the Airborne Snow Observatory (ASO) (Painter et al., 2016). Lidar setup and processing standards were similar to those in the ASO campaigns in California. The data was acquired over the Dischma area near Davos in the eastern Swiss Alps (cf. Figure 3a in Helbig et al., 2020). We used ALS-derived $HS$ maps at three points in time during one winter season, namely at approximate time of peak of winter at 20 March ('PoW') and during early and late-ablation season at 31 March and 17 May 2017 ('abl'). We used a summer DEM from 29 August 2017 to derive the summer terrain parameters.

Each ALS data set covers about 260 km². The original 3 m resolution was aggregated to 5 m horizontal resolution. A RMSE of 13 cm and a bias of -5 cm with snow probing was obtained for within forest but outside canopy (i.e. not below a tree) 1 m ALS-derived $HS$ data from 20 March 2017 (Mazzotti et al., 2019).

### 3.2.3 Terrestrial camera images

We used camera images from terrestrial time-lapse photography in the visible band. The camera (Nikon Coolpix 5900 from 2016 to 2018, Canon EOS 400D from 2019 to 2020) was installed at the SLF/WSL in Davos Dorf in the eastern Swiss Alps (van Herwijnen and Schweizer, 2011; van Herwijnen et al., 2013). Photographs were taken of the Dorfberg in Davos, which is a large southeast-facing slope with a mean slope angle of about 30° (cf. Figure 1 in Helbig et al., 2015a). To obtain $fSCA$ values from the camera images, we followed the workflow described by Portenier et al. (2020). We used the algorithm of Salvatori et al. (2011) to classify pixels in the images as snow or snow free. Though images are taken at regular intervals (between 2 and 15 minutes, depending on the year), we used the image at noon to derive $fSCA$ for that day. We evaluated images from five winter seasons (2016, 2017, 2018, 2019 and 2020) each time from 1 November until 30 June.

The resulting snow/no snow map of the camera images has a horizontal resolution of 2 m. The field of view (FOV) overlaps the most with four 1 x 1 km JIM$_{\mathrm{OSHD}}$ grid cells with projected visible fractions between 9 to 40 % in each grid cell. The camera data set can thus cover roughly about 0.76 km² per time step.

### 3.2.4 Sentinel-2 snow products

We used fine-scale snow-covered area maps, which we obtained from the Theia snow collection (Gascoin et al., 2019). The satellite snow products were generated from Sentinel-2 L2A and L2B images. We used Sentinel-2 snow-covered area maps over one winter season starting at 20 December 2017 until 31 August 2018 for Switzerland. We further used Sentinel-2 snow maps over the Dischma area near Davos close to or at the date of the three days when we had ALS-derived $fSCA$ maps available (18 and 28 March and 17 May 2017).

The horizontal resolution of the snow product is 20 m. While the spatial coverage of the Sentinel-2 snow-covered area maps in Switzerland varies every time step Sentinel-2 may cover several thousands of square kilometers per time step. A validation of the Theia snow product with snow depth from AWS, through comparison to snow maps with higher spatial resolution as well as by visual inspection indicated that snow is detected very well though with a tendency to underdetect snow (Gascoin





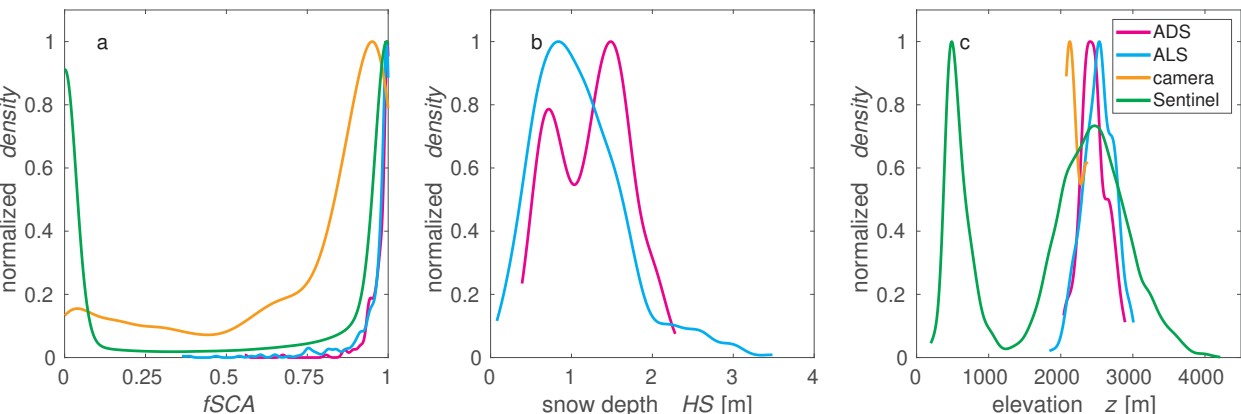

**Figure 2.** Probability density functions after preprocessing for all valid 1 km (a) $fSCA$, (b) $HS$ and (c) elevation $z$ per measurement data set. All densities were normalized with the maximum in each data set. Colors represent the different measurement platforms as detailed in Section 3.2.

et al., 2019). The main difficulty of satellite snow products is to avoid false snow detection within clouds. Furthermore, snow omission errors may occur on steep, shaded slopes when the solar elevation is typically below $20°$.

### 3.3  Derivation of 1 km $fSCA$ evaluation data

For preprocessing, we masked out forest, rivers, glaciers or buildings in all fine-scale measurement data sets. Optical snow products that were obscured by clouds were also neglected. In all fine-scale $HS$ data sets, we neglected $HS$ values that were lower than zero or above 15 m. We used a $HS$ threshold of zero m to decide whether or not a 2 or 5 m grid cell was snow-covered. This threshold could not be better adjusted due to a lack of independent spatial observations. This likely led to the

rather narrow $fSCA$ peak of the probability density function (pdf) around one (cf. pink and light blue line in Figure 2).

We then aggregated all fine-scale snow data as well as the snow products from optical imagery in squared domain sizes $L$ in regular grids of 1 km aligned with the OSHD model domain. For building the spatial averages, we required at least 70 % valid data for the fine-scale snow data and at least 50 % valid for the satellite-derived $fSCA$ data in a domain size $L$ of 1 km. We excluded 1 km domains with spatial mean slope angles larger than $60°$ and spatial mean measured $HS$ lower than 5 cm.

We further neglected 1 km grid cells with forest fractions larger than 10 %, which were derived from 25 m forest cover data. Overall, this led to a varying number of available domains in the different data sets (Table 1). For the fine-scale snow data sets this number ranged from 69 to 157 available valid 1 km domains depending on the point in time with a total of 669 valid 1 km domains. After the removal of clouds and forest we obtained on average every second day in Switzerland some valid Sentinel-2 data (153 valid days from the 255 days). For the time period from 20 December until 31 August 2018, this resulted in 274'979

valid 1 km domains from a total of 3'147'465 valid OSHD grid cells in Switzerland, i.e. about 9 %. These valid 1 km domains cover terrain elevations between 174 m and 4213 m, slope angles between $0°$ to $52°$ and all terrain aspects. We used three of the four grid cells covered by the FOV of the terrestrial camera, since one grid cell had a 1 km forest fraction larger than 10





**Table 1.** Details of the 1 km $fSCA$ evaluation data sets after pre-processing.

| geographical region | remote sensing method | spatial resolution (fine-scale) | spatial coverage | temporal coverage | $\sigma_{fSCA}$ | mean $fSCA$ |
|---|---|---|---|---|---|---|
| | | [m] | [km$^2$] | [days] | | |
| Wannengrat and Dischma area (eastern CH) | ADS | 2 | 232 | 3 | 0.05 | 0.98 |
| Dischma and Engadin area (eastern CH) | ALS | 3 | 437 | 3 | 0.08 | 0.96 |
| Davos Dorfberg (eastern CH) | Terrestrial camera | 2 | 1'019 | 340 | 0.30 | 0.75 |
| Switzerland | Sentinel-2 | 20 | 274'979 | 153 | 0.46 | 0.54 |

%. On average we obtained every fourth day valid camera data (340 valid days from 1211 days). Valid camera-derived $fSCA$ for five seasons and the three grid cells covered by the FOV resulted in 1'019 valid 1 km grid cells from a total of 3'633 1 km grid cells for the five seasons and three grid cells, i.e. 28 %. Compared to the total of all valid OSHD grid cells in Switzerland for the five seasons, the fraction of valid camera-derived $fSCA$ is however less than 0.01 %. The three grid cells have terrain elevations of 2077 m, 2168 m and 2367 m and slope angles of 27°, 34° and 39°. The diversity in each of the evaluation data sets after preprocessing is indicated in Table 1 and is also shown for valid 1 km domains by means of the pdf for $fSCA$, $HS$ and terrain elevation $z$ in Figure 2.

## 3.4 Performance measures

We evaluate modelled and measured $fSCA$ with the following measures: the root mean square error (RMSE), normalized root mean square error (NRMSE, normalized by the mean of the measurements), mean absolute error (MAE) and the mean percentage error (MPE, bias with measured minus modelled and normalized with measurements). We also verify distribution differences by deriving the two-sample Kolmogorov-Smirnov test (K-S test) statistic values $D$ (Yakir, 2013) for the probability density functions (pdf) and by computing the NRMSE for Quantile-Quantile plots (NRMSE$_{quant}$, normalized by the mean of the measured quantiles) for probabilities with values in $[0.1, 0.9]$.

## 4 Results

We grouped the evaluation results of the seasonal $fSCA$ algorithm in three sections: evaluation with $fSCA$ derived from fine-scale $HS$ maps, evaluation with $fSCA$ from time-lapse photography and evaluation with $fSCA$ from Sentinel-2 snow products.

### 4.1 Evaluation with $fSCA$ from fine-scale $HS$ maps

Modelled $fSCA$ compares very well to $fSCA$ derived from all six fine-scale $HS$ data sets. For instance for all evaluated points in time we obtain a NRMSE of 9 % and a MPE of 1 % (Table 2). Overall best performances are achieved for the



**Table 2.** Performance measures are shown for modelled $fSCA$ with (I) $fSCA$ derived from all fine-scale $HS$ maps (combined ADS- and ALS-derived $fSCA$) and (II) Sentinel-derived $fSCA$ (only available for ALS dates). Performance measures are shown for ALS-derived $fSCA$ with Sentinel-derived $fSCA$ (III). Given statistics are NRMSE, RMSE, MPE, MAE, K-S test statistic and NRMSE$_{quant}$. For all differences we computed measured minus modelled values respectively Sentinel-derived $fSCA$ minus ALS-derived $fSCA$ for III. The abbreviations 'acc', 'PoW' and 'abl' indicate the different point in time of the season as given in Section 3.2.

| | NRMSE | RMSE | MPE | MAE | K-S | NRMSE$_{quant}$ |
|---|---|---|---|---|---|---|
| | [%] | | [%] | | | [%] |
| **I JIM$_{OSHD}$ vs ADS&ALS** | | | | | | |
| $fSCA$ | 8.5 | 0.08 | 1.2 | 0.04 | 0.27 | 1.0 |
| $fSCA_{acc}$ | 8.0 | 0.08 | -3.6 | 0.04 | 0.46 | 3.2 |
| $fSCA_{PoW}$ | 4.9 | 0.05 | 0.6 | 0.02 | 0.50 | 0.7 |
| $fSCA_{abl}$ | 10.4 | 0.10 | 2.4 | 0.05 | 0.20 | 2.6 |
| **II JIM$_{OSHD}$ vs Sentinel-2 (at ALS dates)** | | | | | | |
| $fSCA$ | 10.1 | 0.09 | -0.5 | 0.05 | 0.24 | 2.9 |
| $fSCA_{PoW}$ | 2.8 | 0.03 | 2.5 | 0.03 | 1 | 2.7 |
| $fSCA_{abl}$ | 10.2 | 0.09 | -0.6 | 0.05 | 0.22 | 2.9 |
| **III Sentinel-2 vs ALS** | | | | | | |
| $fSCA$ | 10.8 | 0.10 | 3.1 | 0.05 | 0.10 | 4.6 |
| $fSCA_{PoW}$ | 8.7 | 0.08 | -5.9 | 0.06 | 1 | 7.7 |
| $fSCA_{abl}$ | 10.9 | 0.10 | 3.4 | 0.05 | 0.11 | 4.8 |

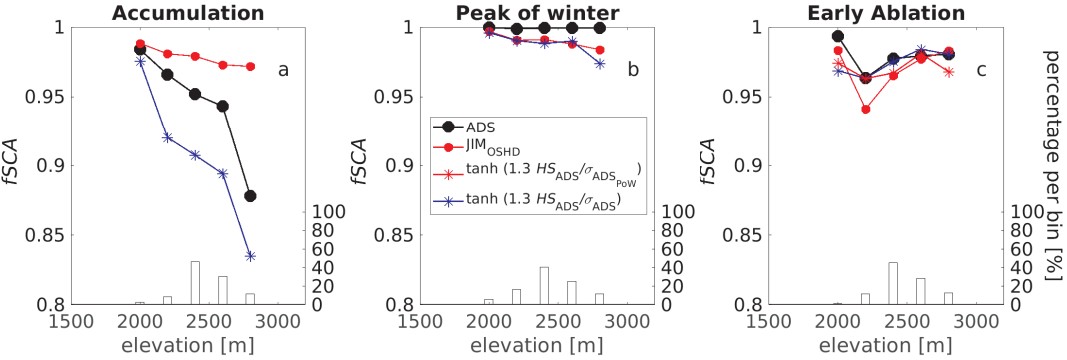

**Figure 3.** Modelled $fSCA$ (JIM$_{OSHD}$) and ADS-derived $fSCA$ in elevation bins for three dates: (a) during accumulation, (b) at approximate peak of winter (PoW) and (c) during ablation. Two benchmarks are shown where applicable. The red stars were derived using Eq. (1) with current ADS $HS$ in the numerator and ADS $\sigma_{HS}$ from the PoW measurement in the denominator. The blue stars were derived using Eq. (1) with current ADS $HS$ in the numerator and current ADS $\sigma_{HS}$ in the denominator. The bars show the valid data percentage per bin.

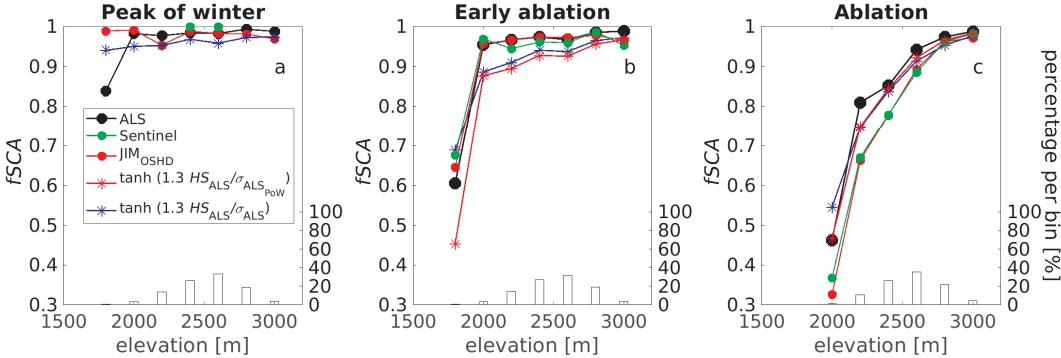

**Figure 4.** Modelled $fSCA$ (JIM$_{OSHD}$), ALS-derived $fSCA$ and Sentinel-derived $fSCA$ in elevation bins for three dates: (a) at approximate PoW, (b) during early ablation and (c) during late ablation. The same two benchmarks as indicated in Figure 3 are shown where applicable. Sentinel-derived $fSCA$ was available 2 days before the PoW, 3 days before the early ablation and at the point in time of the late ablation ALS flight date (green line). The bars show the valid data percentage per bin.

combined two dates at the approximate date of PoW with a NRMSE of 5 % and a MPE of 0.6 %. The performance decreases

slightly for the accumulation date (NRMSE of 8 %) and the combined three points in time of ablation (NRMSE of 10 %).

Given the overall good seasonal agreement between $fSCA$ from all fine-scale $HS$ data sets and modelled $fSCA$, we binned the data in 200 m elevation bands and for ADS and ALS data sets separately to unveil seasonal variations in the elevation-dependent performances. Similar to overall seasonal model performances (Table 2, I), seasonal elevation-dependent performances with ALS data decrease from PoW, to ablation. For ADS data, seasonal elevation-dependent performances are

similar good at PoW and early ablation and decrease during accumulation. Except for the date during accumulation, largest performance differences occur mostly for the lowest elevation bin, i.e. in general, model performances improve with elevation. While at both early ablation dates there is still an overall good agreement between $HS$-derived $fSCA$ and modelled $fSCA$ (red versus black dots in Figure 3c and 4b), at the ablation date modelled $fSCA$ underestimates ALS-derived $fSCA$ across all elevations (Figure 4c). The largest underestimations occur for the two lowest elevation bins with each on average 0.14. Across

all elevations, we obtain almost consistently good performances at approximate PoW (Figure 3b and 4a). Larger overestimations occur only at lowest elevations between 1700 m and 1900 m with on average 0.15. At the date during accumulation, performances decrease with elevation. Modelled $fSCA$ overestimates ADS-derived $fSCA$ at elevations above 2100 m with at maximum 0.09 (Figure 3a).

Some valid Sentinel-2 coverage is available at or close to the dates of the ALS measurements. Though overall seasonal

performances between modelled and Sentinel-derived $fSCA$ decrease from PoW to the combined two ablation dates (Table 2, II), seasonal elevation-dependent performances are best across all elevations for the latest ablation date when Sentinel-2 coverage is available at the exact same day (green versus red dots in Figure 4). At the lowest binned elevations between 1700 m and 1900 m and between 1900 m and 2100 m modelled $fSCA$ underestimates Sentinel-derived $fSCA$ with on average respectively 0.03 and 0.04 (Figure 4b and c). Seasonal performances between Sentinel- and ALS-derived $fSCA$ across all





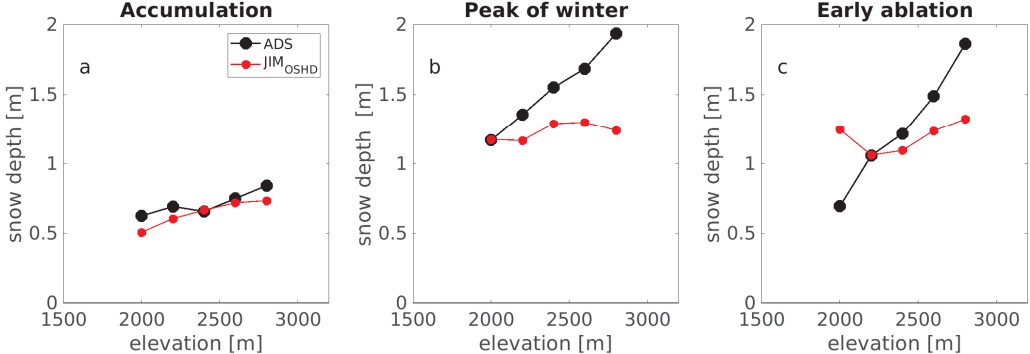

**Figure 5.** Modelled snow depth $HS$ (JIM$_{OSHD}$) and ADS-derived $HS$ in elevation bins for three dates: (a) during accumulation, (b) at approximate PoW and (c) during ablation.

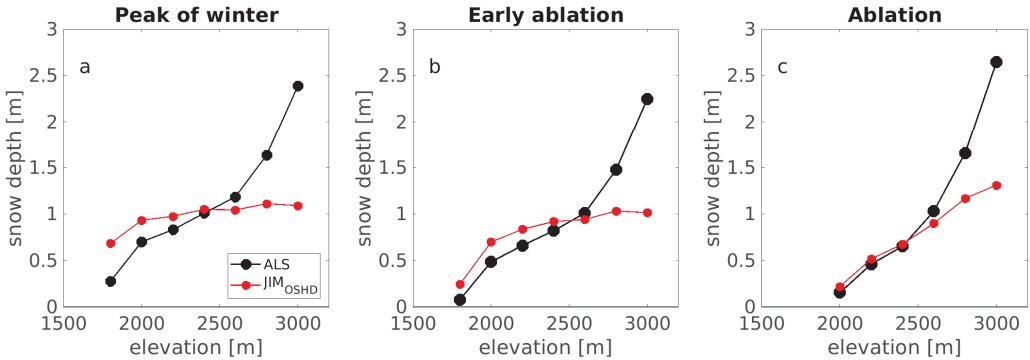

**Figure 6.** Modelled snow depth $HS$ (JIM$_{OSHD}$) and ALS-derived $HS$ in elevation bins for three dates: (a) at approximate PoW, (b) during early ablation and (c) during ablation.

elevations are similar to the performances between modelled and ALS-derived $fSCA$. For all dates with Sentinel-2 coverage we obtain similar NRMSE. Between modelled and Sentinel-derived $fSCA$ the NRMSE is 10 % and between Sentinel- and ALS-derived $fSCA$ the NRMSE is 11 % (Table 2, II versus III).

To understand modelled $fSCA$ performances we also evaluated modelled with measured $HS$ in 200 m - elevation bins (see Figure 5 and 6). Compared to the seasonal snow depth change between the three dates of ADS-$HS$ (Figure 5) there is

much less seasonal variation than between the three dates of the ALS-$HS$ data across all elevations (Figure 6). While on the one hand, the time intervals are much smaller between the three dates of the ALS acquisitions (20 March, 31 March, 17 May 2017) compared to the ones of the ADS acquisitions (26 January, 9 March and 20 April 2016), there were also some snowfall events during ablation in 2017. Except for at the date during accumulation performances decrease with elevation starting at elevations of about 2100 m to 2500 m. Modelled $HS$ considerably underestimates measured $HS$ at higher elevations while

at lower elevations modelled $HS$ mostly overestimates measured $HS$, except for the accumulation and PoW date of the ADS





data. Seasonal performances do not show a clear trend, but best performances are achieved during accumulation. For all dates and data sets, modelled $HS$ shows a NRMSE of 12 % and a MPE of 14 % with measured $HS$.

The $fSCA$ algorithm was implemented in a complex operational snow cover model framework (Section 3.1). Uncertainties related to input or model structure may therefore have an impact on modelled $HS$ and thus $fSCA$ performances. We inves-
tigated this by deriving two benchmark $fSCA$ with Eq. (1) using measured $HS$ data only. The first benchmark $fSCA$ uses current observed $\sigma_{HS}$ and measured $HS$, namely a $fSCA_{\mathrm{curr}}^{\mathrm{measured}}$. The second benchmark model combines current measured $HS$ and observed $\sigma_{HS}$ at PoW, namely a $fSCA_{\mathrm{PoW}}^{\mathrm{measured}}$ (cf. blue and red stars in Figure 3 and 4). At PoW, $fSCA_{\mathrm{PoW}}^{\mathrm{measured}}$ and $fSCA_{\mathrm{curr}}^{\mathrm{measured}}$ are the same and $fSCA_{\mathrm{PoW}}^{\mathrm{measured}}$ can only be derived when PoW has passed, i.e. during ablation. Overall perfor-mances of both benchmark $fSCA$ are better (lower NRMSE) compared to modelled $fSCA$. Among all dates, best seasonal
elevation-dependent performances (200 m bins) of $fSCA_{\mathrm{curr}}^{\mathrm{measured}}$ and $fSCA_{\mathrm{PoW}}^{\mathrm{measured}}$ are achieved for two of the ablation dates (red and blue stars in Figure 3c and 4c). Performances mostly improve, similarly to as for modelled $fSCA$, with elevation. For the three ablation dates, we obtain overall similar NRMSE's for both benchmark models. Except for the lowest elevation bin seasonal elevation-dependent performances are also similar among both benchmark models though the performance of $fSCA_{\mathrm{curr}}^{\mathrm{measured}}$ is slightly improved (e.g. Figure 3c or 4b).

## 4.2 Evaluation with $fSCA$ from camera images

The high temporal resolution of daily camera-derived $fSCA$ allows us to evaluate seasonal model performances. Overall, modelled $fSCA$ follows the seasonal trend of camera-derived $fSCA$ for two of the three grid cells throughout almost all seasons well (cf. for two seasons Figure 7a,c,d,f). However, for the grid cell at 2168 m the ablation season starts much later with modelled $fSCA$ compared to camera-derived $fSCA$, and modelled $fSCA$ further overestimates camera-derived $fSCA$
during accumulation (Figure 7b,e).

For all winter seasons 2016 to 2020 and the three grid cells we obtain a NRMSE of 22 % and a MPE of -7 % for modelled $fSCA$ (Table 3, I). However, interannual performances vary considerably as well as performances among the three grid cells. For instance, for all three grid cells, we obtain the overall best performance for the season 2018 with a NRMSE of 15 % and a MPE of -4 % and the worst performances for season 2019 with a NRMSE of 25 % and a MPE of -12 % and season 2020 with
a NRMSE of 23 % and a MPE of -17 %.

For winter season 2018, we used Sentinel-derived $fSCA$ to evaluate modelled and camera-derived $fSCA$ (Table 3, II and III; Figure 7d,e,f). While modelled and Sentinel-derived $fSCA$ agree very well (NRMSE of 2 % and MPE of -1 %), Sentinel-and camera-derived $fSCA$ compare less well (NRMSE of 12 % and MPE of -5 %) though performances are similar to those for camera-derived and modelled $fSCA$ (NRMSE of 15 % and a MPE of -4 %).

We exploited the high temporal resolution of camera-derived $fSCA$ to evaluate the relevance of applying the full seasonal $fSCA$ algorithm as opposed to snow cover model simplifications of the $fSCA$ algorithm, namely $fSCA_{\mathrm{season}}$ and $fSCA_{\mathrm{curr}}$ (JIM$_{\mathrm{OSHD}}^{\mathrm{season}}$ and JIM$_{\mathrm{OSHD}}^{\mathrm{curr}}$). While $fSCA_{\mathrm{season}}$ and modelled $fSCA$ agree well when the snow cover is quite homogeneous, after snowfalls on partly snow-free ground, $fSCA_{\mathrm{season}}$ can be considerably lower (yellow stars versus red dots in Figure 7b,c). When replacing the $fSCA$ algorithm with $fSCA_{\mathrm{curr}}$, deviations to modelled $fSCA$ using the full algorithm are getting larger



**Table 3.** Performance measures are shown for modelled $fSCA$ and the three grid cells with (I) camera-retrieved $fSCA$ for the winter seasons 2016 to 2019 and for winter season 2018 with (II) Sentinel-derived $fSCA$. Performance measured are shown for all three grid cells for camera-derived $fSCA$ with Sentinel-derived $fSCA$. In (I) statistics are also shown for JIM modelled $fSCA$ versions, namely the algorithm component $fSCA_{season}$ as well as a $fSCA_{curr}$, which uses the current $\sigma_{HS}$ with current $HS$ in Eq. (1) modelled with JIM$_{OSHD}$. Given statistics are NRMSE, RMSE, MPE, MAE, K-S test statistic and NRMSE$_{quant}$.

|  | NRMSE | RMSE | MPE | MAE | K-S | NRMSE$_{quant}$ |
|---|---|---|---|---|---|---|
|  | [%] |  | [%] |  |  | [%] |
| **I JIM$_{OSHD}$ vs camera** |  |  |  |  |  |  |
| $fSCA$ | 21.6 | 0.16 | -7.0 | 0.11 | 0.23 | 9.5 |
| $fSCA_{season}$ | 23.3 | 0.17 | -6.5 | 0.11 | 0.23 | 8.9 |
| $fSCA_{curr}$ | 27.9 | 0.21 | -8.1 | 0.13 | 0.32 | 18.6 |
| **II JIM$_{OSHD}$ vs Sentinel-2** |  |  |  |  |  |  |
| $fSCA$ | 1.8 | 0.02 | -0.7 | 0.01 | 0.53 | 1.03 |
| **III Sentinel-2 vs camera** |  |  |  |  |  |  |
| $fSCA$ | 11.5 | 0.11 | 5.0 | 0.06 | 0.57 | 6.5 |

(blue stars versus red dots in Figure 7). Large overestimations occur similarly after snowfall but large differences now also occur independent from snowfalls during ablation periods. The start of ablation season is delayed but is followed by a much steeper melt out compared to the full $fSCA$ model. Applying $fSCA_{curr}$ always considerably shortens the season compared to applying the full $fSCA$ algorithm. For instance, for season 2016 the shortening is 46 days at 2077 m. In part, $fSCA_{season}$ also shortened the ablation season compared to the full $fSCA$ algorithm by at maximum 24 days at 2077 m in season 2016 [not

shown]. In season 2017 and 2020 however, applying $fSCA_{season}$ prolonged the season by at maximum 6 days at 2168 m in season 2020. Overall, both simplified $fSCA$ models compare less well to camera-derived $fSCA$ than modelled $fSCA$ using the full $fSCA$ algorithm, however $fSCA_{season}$ performs better than $fSCA_{curr}$ (Table 3, I).

### 4.3 Evaluation with $fSCA$ from Sentinel-2 snow products

Overall, modelled $fSCA$ compares well to Sentinel-derived $fSCA$ throughout the season, though there is some elevation-

dependent scatter between modelled and Sentinel-derived $fSCA$ (Figure 8).

In order to analyze the elevation-dependent scatter between modelled and Sentinel-derived $fSCA$, we derived spatial mean $HS$ (solid curve in Figure 8). From this we estimated the end of spatial mean accumulation and the start of spatial mean ablation period for Switzerland at 1 April 2018 (vertical dashed black line in Figure 8). Until the start of the ablation period we obtain the most scatter between modelled and Sentinel-derived $fSCA$ at elevations lower than 1500 m, whereas at higher elevations

both $fSCA$ agree well. At 30 June about 15 % of the seasonal maximum spatial mean $HS$ is left which concentrates at high elevations above about 2700 m (vertical line with stars in Figure 8). From 30 June 2018 until 30 August, i.e. during summer,

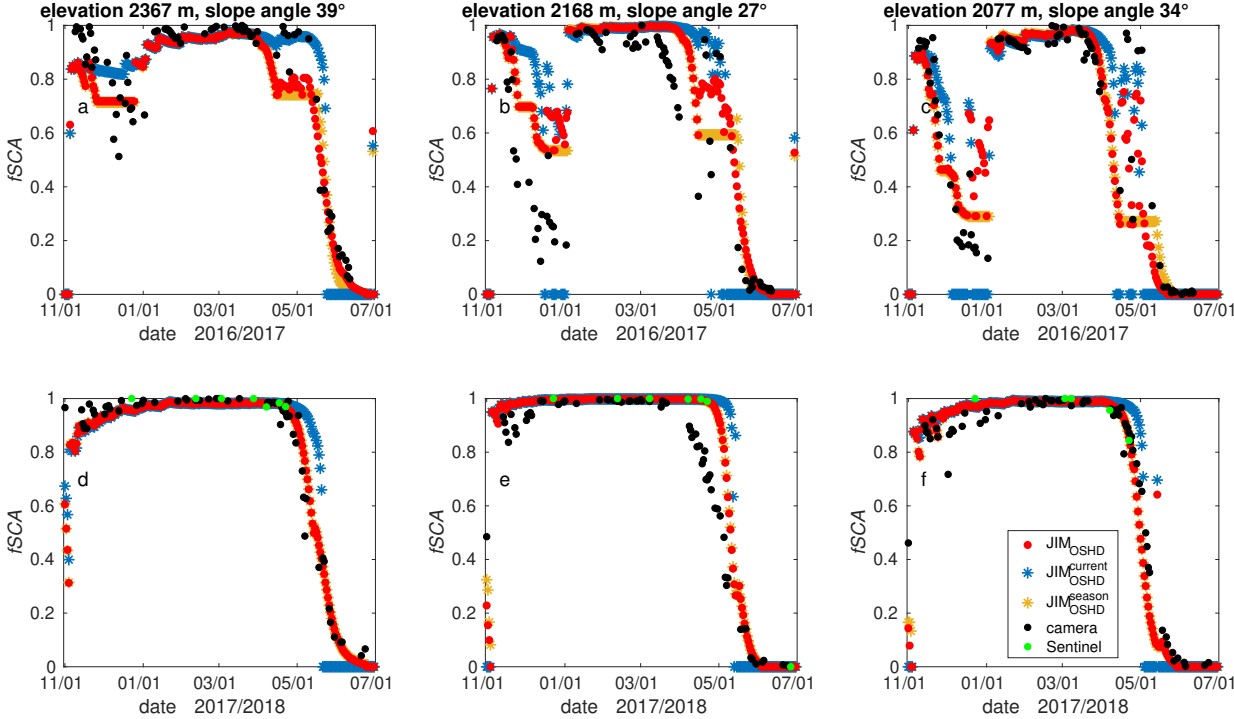

**Figure 7.** Modelled $fSCA$, $fSCA_{season}$, $fSCA_{curr}$ as well as camera-derived $fSCA$ and Sentinel-derived $fSCA$ for the three 1 km grid cells seen by the camera in Davos for two seasons: upper panel (a), (b) and (c) winter 2017, lower panel (c), (d) and (e) winter 2018.

modelled $fSCA$ overestimates Sentinel-derived $fSCA$ at the highest elevations above about 3500 m whereas between snow line and these highest elevations modelled $fSCA$ underestimates Sentinel-derived $fSCA$.

For the winter season lasting from 20 December to 30 June 2018 in Switzerland we obtain a NRMSE of 20 % and a MPE of
2 % (Table 4).

Given the also rather high temporal resolution of the Sentinel-derived $fSCA$ data set, we again computed the $fSCA$ model simplifications, $fSCA_{season}$ and $fSCA_{curr}$. Overall errors with Sentinel-derived $fSCA$ are only slightly worse than for modelled $fSCA$ using the full $fSCA$ algorithm. We obtain a NRMSE of 20 % for $fSCA_{season}$ and a NRMSE of 22 % for $fSCA_{curr}$ (Table 4).

# 5   Discussion

## 5.1   Fractional snow-covered area $fSCA$ algorithm

We developed a seasonal $fSCA$ algorithm by combining a PoW $\sigma_{HS}$ parameterization for mountainous terrain (Eq. (2)) and one for flat terrain (Eq. (3)) with tracking snow values for alternating accumulation and melt events throughout the season in a closed form $fSCA$ parameterization (Eq. (1). Such an implementation of a seasonal $fSCA$ algorithm has, to the best of



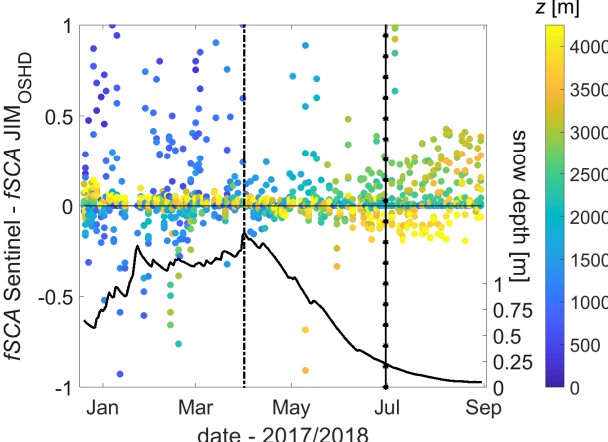

**Figure 8.** Sentinel-derived $fSCA$ minus modelled $fSCA$ for Switzerland as a function of date and elevation $z$ for available satellite dates. Daily spatial mean snow depth $HS$ is shown by the solid line below. Approximate end of accumulation and start of ablation season is indicated by the dashed vertical line whereas the approximate end of ablation season is indicated by the vertical line with stars.

**Table 4.** Performance measures between Sentinel-derived $fSCA$ and modelled $fSCA$ for all valid 1 km grid cells of Switzerland between 20 December 2017 and 30 June 2018. Given statistics are NRMSE, RMSE, MPE, MAE, K-S test statistic and NRMSE$_\text{quant}$.

|  | NRMSE | RMSE | MPE | MAE | K-S | NRMSE$_\text{quant}$ |
|---|---|---|---|---|---|---|
|  | [%] |  | [%] |  |  | [%] |
|  |  |  |  |  |  |  |
| $fSCA$ | 19.9 | 0.15 | 1.9 | 0.05 | 0.39 | 2.5 |
| $fSCA_\text{season}$ | 20.1 | 0.15 | 1.9 | 0.05 | 0.39 | 2.6 |
| $fSCA_\text{curr}$ | 22.0 | 0.16 | 1.1 | 0.06 | 0.39 | 4.5 |

our knowledge, not been presented in detail so far. The algorithm is easy to apply and only requires storing snow history and subgrid summer terrain parameters, which are the slope related parameter $\mu$ and the terrain correlation length (Section 2.2).

At the moment we use the $\sigma_{HS}^{\text{flat}}$ parameterization (Eq. (3)) to describe the spatial new snow depth distribution $\sigma_{HS}$ in Eq. (1) rather than the $\sigma_{HS}^{\text{topo}}$ parameterization (Eq. (2)). Since $\sigma_{HS}^{\text{topo}}$ was empirically derived from PoW data we found that to describe the spatial new snow depth distributions in mountainous terrain when the ground is typically almost completely

covered by snow we might need a different description. As a first approach we therefore use the flat field parameterization even over mountainous terrain. Though at least at lower elevations and during spring neglecting topographic interactions might be justified for new snow distributions, spatial snow depth distributions before and after snowfall accumulations should be analyzed throughout the season for confirmation.





Implementing the seasonal $fSCA$ algorithm in a distributed snow cover model allowed us to evaluate the algorithm with

spatiotemporal measurement data. We are not aware of any seasonal $fSCA$ implementation that has been evaluated in detail

by exploiting independent $HS$ data sets in high spatial resolution and snow products in high temporal resolution.

## 5.2 Evaluation

### 5.2.1 Evaluation with $fSCA$ from fine-scale $HS$ maps

The evaluation of the seasonal $fSCA$ algorithm with $fSCA$ from fine-scale $HS$ maps revealed overall good performances

at all six points of the season with NRMSE's always being lower than 10 % (Table 2). Performances decreased from PoW, to

accumulation and later ablation.

During accumulation at higher elevations modelled $fSCA$ overestimates ADS-derived $fSCA$ though modelled $HS$ under-

estimates measured $HS$ across all elevations (Figure 3a and 5a). This could indicate a problem of our $fSCA$ algorithm during

accumulation. In this period of the season snowfall events dominate, during which, we use the flat field standard deviation

of $HS$ (Eq. (3)) to characterize $fSCA$ even on inclined grid cells. Not accounting for the various topography interactions

with wind, precipitation and radiation shaping the snow depth distribution in mountainous terrain during accumulation might

have led to overestimations of modelled $fSCA$. The description of spatial $HS$ distribution during accumulation thus requires

further investigations, for which however more than one spatial $HS$ data set acquired during accumulation would be needed.

Except for during accumulation, modelled $fSCA$ rather underestimates $fSCA$ from fine-scale $HS$ maps. However, mod-

elled $fSCA$ does not show similar strong trends when compared to Sentinel-derived $fSCA$ but agrees rather well with $fSCA$

from Sentinel-2 snow products for the three dates (Figure 4). Largest underestimations occur for ALS data at lower elevations

and during ablation where low $HS$ values of on average lower than 30 cm dominate (Figure 6). We assume that the choice of a

$HS$ threshold of zero m to decide whether or not a 2 or 5 m grid cell was snow-covered might be one reason for the underesti-

mations. In reality small positive or negative $HS$ values might have been zero too. When increasing this threshold to $\pm$ 10 cm

resulting 1 km $fSCA$ from $HS$ maps decreased considerably and in part large overestimations of modelled $fSCA$ resulted

at the various points in time of the season [not shown]. Unfortunately, we currently do not have detailed snow observations

available to define robust $HS$ threshold values which take into account the different points in time of the season as well as

varying terrain slope angles. However, the overall good agreement between Sentinel- and ALS-derived $fSCA$ (Figure 4 and

Table 2, III) provides confidence in the fine-scale $HS$ data-derived $fSCA$ used here to evaluate modelled $fSCA$.

$fSCA$ performances mostly improve with elevation or remain similar, except for during accumulation (Figure 3b,c and 4).

On the contrary, performances for modelled $HS$ mostly decrease with elevation for the same points in time (Figure 5b,c and

6). Large underestimations in modelled $HS$ at high elevations affected modelled $fSCA$ much less than weak overestimations

of measured $HS$ at lower elevation during ablation. This is not contradictory but emphasizes the need of accurately modelled

$HS$ along snow lines where small inaccuracies in $HS$ can have large impacts. In addition, along the snow line the valid data

percentage per bin was very low with values between 1 to 5 % for all $fSCA$ from fine-scale $HS$ data sets. Thus, a single outlier

along the snow line could have also degraded the performance (e.g. Figure 5c). Note that the overall tendency of modelled $HS$





to underestimate measured $HS$ at high altitudes may also originate from precipitation underestimation. As there are fewer AWS at high elevations data assimilation cannot correct for any flawed precipitation input.

The two benchmark $fSCA$ models ($fSCA_{\mathrm{curr}}^{\mathrm{measured}}$ and $fSCA_{\mathrm{PoW}}^{\mathrm{measured}}$) using measured $HS$ compare better to $fSCA$ derived
from $HS$ data than modelled $fSCA$ using JIM$_{\mathrm{OSHD}}$. This result confirms the previously derived functional $tanh$-form (Eq. (1)) for $fSCA$ at PoW for a seasonal application. While at the date of early ablation of ALS data, modelled $fSCA$ performed better, this might be due to snowfalls after the date at approximate PoW with consecutive melt (Figure 4b). This may have altered the actual PoW snow depth distribution compared to the ALS-measured $\sigma_{HS}$ at approximate date of PoW. Except for the lowest elevation bin, performances among both benchmark models are quite similar. While we would have expected at least
a better performance of $fSCA_{\mathrm{PoW}}^{\mathrm{measured}}$ during ablation, $fSCA_{\mathrm{curr}}^{\mathrm{measured}}$ performs slightly better during early ablation. The reason for this is most likely the same than why modelled $fSCA$ outperformed both benchmark models at that early ablation date (Figure 4b). Due to snowfalls after the approximate date of PoW of ALS data, at some elevations, the actual PoW snow depth distribution does not agree with the one at approximate date of PoW of ALS data at these elevations anymore. Applying a snow cover model that tracks the history of $HS$ to derive seasonal $fSCA$ is thus beneficial. Evaluating the benchmark $fSCA$
models with $fSCA$ derived from $HS$ data confirmed the overall applicability of our seasonal $fSCA$ algorithm.

### 5.2.2 Evaluation with camera-derived $fSCA$

While the evaluation of the seasonal $fSCA$ algorithm with $fSCA$ from fine-scale $HS$ maps revealed overall good performances at six points in time, seasonal performances could not be evaluated continuously over the season. Evaluating with daily camera-derived $fSCA$ demonstrated that modelled $fSCA$ was able to mostly reproduce well the seasonal trend (Figure 7).
However, overall, modelled $fSCA$ compared less well to camera-derived $fSCA$ than modelled $fSCA$ compared to $HS$-derived $fSCA$ (e.g. NRMSE of 22 % compared to NRMSE to 9 %; Table 2, I versus Table 2, I). These overall larger errors most likely originate in an overall lower accuracy of camera-derived $fSCA$ compared to $fSCA$ from fine-scale $HS$ maps. For instance, the projection of the 2D-camera image to a 3D DEM may introduce errors and distortions. Furthermore, when deriving $fSCA$ from camera images, clouds/fog and uneven illumination due to for instance shading or partial cloud cover
may compromise the possibility of detecting snow by the snow classification algorithm of Salvatori et al. (2011) and can deteriorate the accuracy (e.g. Farinotti et al., 2010; Fedorov et al., 2016; Härer et al., 2016; Portenier et al., 2020). The choice of the threshold method when automatically deriving $fSCA$ from the images also introduces uncertainty. Here, we decided that the method proposed by Salvatori et al. (2011) followed the seasonal modelled $fSCA$ trend best though some uncertainty remained. For instance, the decreased performances by about 10 % of the NRMSE in season 2019 and 2020 could stem from an
increase in the number of image pixels when the camera was upgraded. This may have led to more detailed information when e.g. small vegetation is resolved. The overall better agreement between modelled and Sentinel-derived $fSCA$ than between between Sentinel- and camera-derived $fSCA$ (NRMSE of 2 % versus 12 %, cf. Table 2) similarly indicates some larger uncertainties in the camera-derived $fSCA$ data set. For instance, while we required at least 50 % valid fine-scale information for the Sentinel-derived $fSCA$ when aggregating to 1 km $fSCA$ maps, this requirement could not be met for camera-derived
$fSCA$. For the three 1 km model grid cells the projected fractions of the camera FOV are 9 %, 13 % and 14 %, which is





much lower than the 50 % but is also used to evaluate modelled $fSCA$ for the full grid cell area. On the other hand, while it seems that there is a better agreement between Sentinel-derived and modelled $fSCA$ than between camera-derived and modelled $fSCA$, valid Sentinel-derived $fCSA$ has a much lower temporal resolution and did not cover the entire ablation period. Instead, Sentinel-derived $fSCA$ was often available throughout the period when $fSCA$ was rather close to one (cf.

Figure 7d,e). Thus, while there is likely more uncertainty in camera-derived $fSCA$, the snow cover model might have also underestimated snow melt which led to overestimated modelled $HS$ and thus $fSCA$ at the beginning of ablation (cf. Figure 7e).

The high temporal resolution of camera-derived $fSCA$ allowed us to evaluate modelled simplifications of the seasonal $fSCA$ algorithm, i.e. $fSCA_{\text{season}}$ and $fSCA_{\text{curr}}$ (JIM$_{\text{OSHD}}^{\text{season}}$ and JIM$_{\text{OSHD}}^{\text{curr}}$). While the overall performance decrease is rather

low with for instance an increase in NRMSE by 1 % for JIM$_{\text{OSHD}}^{\text{season}}$ and by 6 % for JIM$_{\text{OSHD}}^{\text{curr}}$ compared to the full $fSCA$ model, seasonal performance trends are clearly poorer than when applying the full $fSCA$ model (Figure 7). The reason that this deterioration is not seen in the overall error measures is most likely due to less frequent camera-derived $fSCA$ at time steps during or following snowfall events when clouds or bad illumination might have prevented deriving valid $fSCA$ from images. While the in part large overestimations of camera-derived $fSCA$ increase from JIM$_{\text{OSHD}}^{\text{season}}$ to JIM$_{\text{OSHD}}^{\text{curr}}$, with JIM$_{\text{OSHD}}^{\text{curr}}$ the start

of the ablation season is not only delayed but the ablation season is also considerably shortened by up to 46 days. In principle, $fSCA_{\text{curr}}$ describes seasonal $fSCA$ as if staying continuously at peak winter, though for various $HS$ values. However, this leads to sudden jumps when current $HS$ approaches zero, as seen by the steep melt outs of JIM$_{\text{OSHD}}^{\text{curr}}$, or when current $HS$ raises from no snow to a value larger than zero following snowfall events on bare ground, as seen during accumulation for JIM$_{\text{OSHD}}^{\text{curr}}$. Thus, while including the tracking of current seasonal maximum $HS$ to derive the current maximum $\sigma_{HS}$ already improved

the seasonal trends ($fSCA_{\text{season}}$), additional accounting for $fSCA_{\text{nsnow}}$ is able to overcome the remaining differences between $fSCA_{\text{season}}$ and modelled $fSCA$ derived by the full $fSCA$ algorithm.

### 5.2.3 Evaluation with Sentinel-derived $fSCA$

By including Sentinel-derived $fSCA$ in our evaluation data set to evaluate modelled $fSCA$, we added a data set that unites a rather high temporal data resolution with a much larger spatial coverage than was inherent in the two other evaluation data

sets (cf. Table 1). The Sentinel-derived $fSCA$ data set comprises about 275'000 1 km grid cells covering a range in terrain elevations, slope angles and terrain aspects. This variety was not achieved for the high-temporal evaluation with camera-derived $fSCA$ limited to one southeast-facing slope with overall similar elevations between 2077 m and 2367 m and slope angles between 27° and 39° (cf. Figure 2b).

For the one winter season investigated, we obtained an overall good seasonal agreement across Switzerland, though some

elevation-dependent scatter exists (Figure 8). The majority of the largest scatter occurs during the accumulation period at lower elevations where lower spatial $HS$ values as well as more cloudy weather prevail during accumulation. By neglecting all 1 km domains with modelled $HS$ lower than 5 cm, which would also resemble the preprocessing of fine-scale $HS$-derived $fSCA$ (cf. Section 3.3), the scatter between modelled and Sentinel-derived $fSCA$ at these lower elevations during accumulation reduced considerably and the overall performances improved substantially. For instance the NRMSE reduced from 20 % to 12





% and the MPE from 1.9 % to 0.23 %. The scatter at higher elevations during summer might originate from underestimated modelled $fSCA$ due to underestimated precipitation (fewer AWS at high elevations).

    Similar than for camera-derived $fSCA$ the overall performance decrease when using $JIM_{OSHD}^{season}$ and $JIM_{OSHD}^{curr}$ is rather low with for instance an increase in NRMSE by 0.2 % for $JIM_{OSHD}^{season}$ and by 2 % for $JIM_{OSHD}^{curr}$ compared to the full $fSCA$ model. When binned per elevation for Switzerland a small increase in scatter only appeared between modelled $fSCA$ and $fSCA_{curr}$

towards the end of the season [not shown]. While we in part obtained large differences for individual grid cells between the three modelled $fSCA$ and camera-derived $fSCA$, performances between modelled and Sentinel-derived $fSCA$ only improved slightly compared to when applying $JIM_{OSHD}^{season}$ or $JIM_{OSHD}^{curr}$ over a much larger spatial coverage. We assume that the lack of a stronger improvement in the overall error measures is due to more missing valid satellite coverage during clouded periods that typically occur during or after snowfalls. Yet exactly during these periods we would expect larger differences due

to the missing new snow $fSCA$ updates when e.g. reducing the full $fSCA$ model to $fSCA^{season}$ (cf. Figure 7b,c). Overall, we obtained poorer performance measures between modelled $fSCA$ and Sentinel- as well as camera-derived $fSCA$ compared to between modelled $fSCA$ and $fSCA$ from fine-scale $HS$ maps (e.g. a NRMSE of 20 % for Sentinel-2 $fSCA$, of 22 % for camera $fSCA$ and of 9 % for $fSCA$ from $HS$ data). Uncertainties introduced by reduced visibility in the snow products of Sentinel-2 and the camera are most likely the reason. Both, our camera- as well as the Sentinel-2 data set cover long time

periods in higher temporal resolution, i.e. they include also periods under unfavorable weather conditions. On the contrary, clear sky dates were carefully selected for the on-demand high-quality data acquisitions from the air for our $fSCA$ data sets derived from fine-scale $HS$ maps. Nevertheless, the camera- as well as the Sentinel-2 data set enabled us to evaluate seasonal $fSCA$ model trends which would not have been possible alone from the six $fSCA$ data sets derived from $HS$ data.

## 6   Conclusions

We presented a seasonal fractional snow-covered area ($fSCA$) algorithm based on the $fSCA$ parameterization of Helbig et al. (2015b, 2020). The seasonal algorithm is based on tracking $HS$ and $SWE$ values accounting for alternating snow accumulation and melt events. Two empirical parameterizations are applied to describe the spatial snow depth distribution, one for mountainous terrain at PoW and one for flat terrain during a snowfall. An implementation in a multilayer energy balance snow cover model system ($JIM_{OSHD}$; JIM, JULES investigation model (Essery et al., 2013)) allowed us to evaluate seasonally

modelled $fSCA$ for Switzerland.

    Compiling independent $fSCA$ data sets enabled a thorough spatiotemporal analysis of the seasonal $fSCA$ algorithm. While the evaluation with the three data sets showed overall good seasonal performance, each of the evaluation data sets allowed to draw additional conclusions. The evaluation with fine-scale spatial $HS$-derived $fSCA$ showed that snow depth uncertainties along the snow line likely contributed to the largest $fSCA$ underestimations during ablation compared to the overall best

agreement at PoW. The camera-derived $fSCA$ data set, with the highest temporal resolution, confirmed the need for tracking $HS$ over the season as well as accounting for intermediate snowfalls to avoid a delayed melt start and a drastically shortening of the ablation season. The Sentinel-derived $fSCA$ data set, with the largest spatial coverage together with a rather high temporal



resolution, demonstrated that the seasonal $fSCA$ algorithm performs well across a range of elevations, slope angles, terrain aspects and snow regimes. This comparison showed that there were some differences at low elevation coinciding with very

low $HS$ early in the season, while discrepancies occured mostly at high elevations towards the end of the season respectively during summer.

Overall NRMSE's for seasonally modelled $fSCA$ increased from 9 % for $HS$ data-derived $fSCA$, to 20 % for Sentinel-derived $fSCA$ and to 22 % for camera-derived $fSCA$. While the large margin in performance measures is likely tied to the various temporal and spatial resolutions of the data sets leading to different data uncertainties, it also demonstrates the diffi-

culties in drawing conclusions when evaluating a model algorithm with evaluation data from different acquisition platforms. Nevertheless, this comparison with data covering a wide range of spatiotemporal scales allowed us to obtain a comprehensive overview of the strength and weaknesses of our seasonal $fSCA$ implementation.

The implementation of the seasonal $fSCA$ algorithm in a model only requires tracking $HS$ and $SWE$ for a coarse grid cell as well as deriving subgrid summer terrain parameters from a fine-scale summer DEM. The PoW $fSCA$ parameterization

of Helbig et al. (2020) forms the centerpiece of the presented seasonal $fSCA$ algorithm. The recent evaluation with various spatial PoW snow depth data sets from 7 geographic regions showed an overall NRMSE of only 2 %. This detailed evaluation at PoW in different geographic regions and the seasonal evaluation with the three $fSCA$ data pools presented here, suggests that the seasonal $fSCA$ algorithm may perform similar in most other geographic regions. However, further investigations, once more spatial $HS$ data sets before and after snowfalls in complex topography become available, would be advantageous

for improvements of our seasonal $fSCA$ algorithm during a snowfall.

*Code availability.*    The code of the depletion curve implementation will be made available upon final publication.

*Data availability.*    All data used in this study is described in the data section. The data can be downloaded from the referenced repositories or data availability is described in the referenced publications. Theia snow maps are freely distributed via the Theia portal (https://doi.org/10.24400/329360/F7Q52MNK).

*Competing interests.*    The authors declare that they have no conflict of interest.

*Acknowledgements.*    We thank Andreas Stoffel at SLF for his help with GIS processing of the satellite images. N. Helbig was funded by a grant of the Swiss National Science Foundation (SNF) (Grant N° IZSEZ_186887), as well as partly funded by the Federal Office of the Environment FOEN.





# 1 Appendix: Technical aspects - Seasonal $fSCA$ implementation

The technical aspects of the different $fSCA$ (cf. box in the middle of in Figure 1), i.e. the seasonal $fSCA$ ($fSCA_{\text{season}}$) and the $fSCA$ for snowfall events ($fSCA_{\text{nsnow}}$), are given here. This description gives the necessary details to implement the seasonal $fSCA$ algorithm in a snow cover model. We first present some pseudocode and then give a detailed text description.

!! *Seasonal $fSCA$ algorithm*

```
for each grid cell do
      !! Update SWE history (buffer) from past 14 days with current SWE
      SWE_buffer(current)=SWE
      !! Calculate max, min and recent min indices in 14 days SWE_buffer
      max_buff, min_buff, recentmin_buff
!! Apply indices to finding new snow depth changes ΔHS
      !! New snow amount in 14 days buffer
      14 day ΔHS = HS - HS(min_buff)
      !! Recent new snow amount in 14 days buffer
     recent ΔHS = HS - HS(recentmin_buff)
!! Max snow depth change in 14 days buffer
     max ΔHS = HS(max_buff) - HS(min_buff)
     !! Find current absolute max and pseudo-min SWE values
     IF SWE is zero, set SWE_max and SWE_pseudo-min to zero
     IF SWE ≥ SWE_max, set SWE_max and SWE_pseudo-min to SWE
IF SWE < SWE_max and SWE < SWE_pseudo-min, set SWE_pseudo-min = SWE
     set HS_max, HS_pseudo-min according to SWE_max,SWE_pseudo-min
     !! Start calculating fSCA
     !! fSCA_season using Eq. (1)-(3)
     IF grid cell is flat
σ_HSseason := Eq. (3) with HS_max
     ELSE
     σ_HSseason := Eq. (2) with HS_max
     fSCA_season := Eq. (1) with σ_HSseason and HS_pseudo-min
     !! fSCA_14daynsnow using Eq. (1) and (3)
σ_HS14d := Eq. (3) with max ΔHS
     fSCA_14daynsnow := Eq. (1) with σ_HS14d and 14 day ΔHS
     !! fSCA_recentnsnow using Eq. (1) and (3)
```





| | 29 | $\sigma_{HS\text{recent}}$ := Eq. (3) with recent $\Delta HS$ |
|---|---|---|
| | 30 | $fSCA_{\text{recentnsnow}}$:= Eq. (1) with $\sigma_{HS\text{recent}}$ and recent $\Delta HS$ |
| 530 | 31 | !! *Deriving* $fSCA_{nsnow}$ |
| | 32 | $fSCA_{\text{nsnow}} =\max(fSCA_{\text{14daynsnow}}, fSCA_{\text{recentnsnow}})$ |
| | 33 | !! *Reset* $fSCA_{season}$, *if new snow is really melting* |
| | 34 | IF $fSCA_{\text{nsnow}} > 0$ and $fSCA_{\text{nsnow}} < fSCA_{\text{season}}$ |
| | 35 | $SWE_{\text{pseudo-min}} = SWE$ and $HS_{\text{pseudo-min}} = HS$ |
| 535 | 36 | !! *Calculate coefficient of variation from seasonal values* |
| | 37 | $CV_{\text{season}} = \sigma_{HSseason}/HS_{\text{max}}$ |
| | 38 | !! *Recalculate current absolute* $HS_{max}$ |
| | 39 | $HS_{\text{max}} = 1.3HS_{\text{pseudo-min}}/(CV_{\text{season}}\text{atanh}(fSCA_{\text{season}}))$ |
| | 40 | !! *Recalculate current absolute* $SWE_{max}$ |
| 540 | 41 | $SWE_{\text{max}} = \rho_{\text{max}}HS_{\text{max}}$ |
| | 42 | !! *Recalculate* $fSCA_{season}$ |
| | 43 | IF grid cell is flat |
| | 44 | $\sigma_{HSseason}$ := Eq. (3) with $HS_{\text{max}}$ |
| | 45 | ELSE |
| 545 | 46 | $\sigma_{HSseason}$ := Eq. (2) with $HS_{\text{max}}$ |
| | 47 | $fSCA_{\text{season}}$:= Eq. (1) with $\sigma_{HSseason}$ and $HS_{\text{pseudo-min}}$ |
| | 48 | $fSCA_{\text{nsnow}}$:=0 |
| | 49 | !! *Calculate final* $fSCA$ |
| | 50 | $fSCA=\max(fSCA_{\text{season}}, fSCA_{\text{nsnow}})$ |

Following new snow accumulation, the ground is almost completely covered by snow, which may lead to a different spatial snow depth variability than at PoW. We account for this by using $\sigma_{HS}^{\text{flat}}$ rather than $\sigma_{HS}^{\text{topo}}$ for the derivation of $fSCA_{\text{nsnow}}$ to avoid introducing topography interactions in new snow $\sigma_{HS}$ which were derived for PoW $\sigma_{HS}$. To calculate $fSCA_{\text{nsnow}}$ we insert new snow amounts in Eq. (1)-(3). Thus, $fSCA_{\text{nsnow}}$ describes the contribution to $fSCA$ solely from the new snow, i.e.

as if the new snow fell on bare ground. Two $fSCA_{\text{nsnow}}$ are derived: $fSCA_{\text{14daynsnow}}$ for a new snow event within the last 14 days and a $fSCA_{\text{recentnsnow}}$ for the most recent new snow event. To calculate both, $fSCA_{\text{14daynsnow}}$ and $fSCA_{\text{recentnsnow}}$, we store $HS$ of the last 14 days. For $fSCA_{\text{14daynsnow}}$ we derive the absolute maximum as well as the absolute minimum from this time window. The difference between these two extreme $HS$ values is used to compute the corresponding $\sigma_{HS}$ and the difference between current and absolute minimum $HS$ is inserted in the numerator to obtain $fSCA_{\text{14daynsnow}}$ as $fSCA$.

To compute $fSCA_{\text{recentnsnow}}$ we determine the first local $HS$ minimum from the 14 days time window by going back in time. The difference between current and this local minimum $HS$ is used to derive $\sigma_{HS}$ and is also used in the numerator of $fSCA_{\text{recentnsnow}}$. The maximum of $fSCA_{\text{14daynsnow}}$ and $fSCA_{\text{recentnsnow}}$ gives us $fSCA_{\text{nsnow}}$ for that time step and grid cell.





To describe the overall seasonal $fSCA$ development we use a $fSCA_\text{season}$ which we compute with $\sigma_{HS}^\text{topo}$. For grid cells with slope angles equal to zero we use $\sigma_{HS}^\text{flat}$. To compute $fSCA_\text{season}$ we use current seasonal maximum $HS$ to derive $\sigma_{HS}^\text{topo}$ or $\sigma_{HS}^\text{flat}$. In the numerator of $fSCA_\text{season}$ we use a $HS$ variable which we call a pseudo-minimum $HS$ solely to differentiate it from real global and local minima. The pseudo-minimum $HS$ is used in $fSCA_\text{season}$ to derive a $fSCA$ as if there was no previous snowfall. We do this to obtain two separate $fSCA$, one $fSCA_\text{nsnow}$ and one $fSCA_\text{season}$, which will be compared afterwards. During accumulation, the pseudo-minimum $HS$ is the current $HS$ up until a snow event starts, following a previous melt period. Then the pseudo-minimum $HS$ keeps the pre-snow event $HS$ value up until current $HS$ reaches the current seasonal maximum $HS$ again. From then on the pseudo-minimum $HS$ is the current $HS$ again. During ablation, the pseudo-minimum $HS$ matches, similar as during accumulation, the current $HS$ up until a snow event starts. Then the pseudo-minimum $HS$ keeps the pre-snow event $HS$ value up until current $HS$ falls below the pre-snow $HS$ value again or increases up to a new current seasonal maximum $HS$. However, once the $fSCA_\text{nsnow}$ is again lower than $fSCA_\text{season}$ and the newly fallen snow has started to melt ($SWE_{t-1} - SWE_t > 2$ mm), we recalculate the current seasonal maximum $HS$. Then, we update $fSCA_\text{season}$ using the new current seasonal maximum $HS$ for $\sigma_{HS}$ and the pseudo-minimum $HS$ taking the current $HS$ in the numerator. We perform the recalculation of the seasonal maximum $HS$ to account for an increased seasonal $\sigma_{HS}$ caused by the intermediate snow event. The recalculated seasonal maximum $HS$ takes that value that allows to arrive at the current $HS$ by melt only, i.e. without intermediate snowfall. For the recalculation procedure we solve the seasonal $CV$ from before the snow event, i.e. $\sigma_{HS}/HS$ both using the previous seasonal maximum $HS$, for $\sigma_{HS}$ and insert it in $fSCA_\text{season}$. By further using the pseudo-minimum $HS$ (which was set to the current $HS$) in $fSCA_\text{season}$ we derive a new seasonal maximum $HS$. At the end of this adjustment $fSCA_\text{nsnow}$ is set to zero and an updated (larger) seasonal maximum $HS$ with a similar or slightly lower $fSCA_\text{season}$ results.





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
