# Peer review of "A seasonal algorithm of the snow-covered area fraction for mountainous terrain"

_The Cryosphere, 2020_

## Author Comment (AC1)

We thank the reviewer# 1 for the review and the comments. All comments (in italics) are addressed below (in bold).

**General comments**

*Snow cover has a strong influence on surface energy balance but does not always uniformly blanket the ground. There have therefore been many papers proposing parameterizations for fractional snow-covered area in surface energy balance models, often very simple and based on limited observations. Ideally, a seasonal snow cover parameterization will account for terrain influences, the scale of the model cells and hysteresis between accumulating and melting snow covers. Helbig et al. build on their valuable earlier work to present such a parameterization and evaluate it with several extensive observed datasets. There is good work here, but it is very hard work for the reader; I have read the paper three times and am still struggling. I think that the descriptions, the evaluations and the algorithm itself need to be substantially simplified.*

**Thank you very much for this comment. We went over the manuscript to make it easier to read and have rewritten large parts. In particular, we completely rewrote the description of the algorithm, and now also included two figures to better illustrate how it works. Please see our answers on all issues below.**

*We are referred to Helbig et al. (2015b, 2020) for details of the algorithm, and it is actually impossible to tell what is being done here without reading those papers. Brief explanations of how c, d, µ and ξ are calculated should be given. The appendix will be essential (but not quite sufficient) for anyone wishing to implement this algorithm in another model, and the schematic in Figure 1 should be moved to that appendix (the figure is not fully comprehensible just from material presented in the main text). For readers wanting to get an overview of the method, I suggest that an alternative Figure 1 showing typical modelled $fSCA$ behaviour over a season would be better (this is more or less done in Figure 7, but without explaining why the models differ in the ways that they do).*

**We completely rewrote the description of the $fSCA$ algorithm and also added two new figures to illustrate and better understand our algorithm. Furthermore, we published the algorithm code on a gitlab repository and linked it to EnviDat, an environmental data portal.**

*Six different performance measures are presented with very little consideration of what aspects of performance they measure and a lack of context for what could be considered a good performance. The Kolmogorov-Smirnov test statistic implies a significance test and the Q-Q plot statistic suggests a comparison of distribution shapes that are never presented. Cut this down to a set of measures that are meaningfully used to measure performance and*

*to communicate information.*

We reduced the number of measures to NRMSE, RMSE and MPE, which we discuss in the manuscript.

*The Niu and Yang (2007) fSCA parameterization can be implemented in one line of code and includes hysteresis to some extent through snow density. Just the pseudocode for the algorithm presented here requires 32 non-comment lines and contains many apparently ad hoc design decisions: what is the significance of 14 days for new snow accumulation?how flat does a flat cell have to be? why use the flat parameterization for new snow in mountains rather than any other value? Considering uncertainties in observations revealed when different datasets overlap, errors in the modelled mass balance and ad hoc decisions, is the complexity justified? Tables 3 and 4 and Figure 7 are not very convincing in this regard.*

Indeed, the closed-form $fSCA$ parameterization from Niu and Yang (2007) is a one line code - which is much simpler compared to our seasonal algorithm. However, Niu and Yang (2007) was developed and tested on monthly $fSCA$ data on spatial scales of 1° by 1°. Swenson and Lawrence (2012) demonstrated that this algorithm cannot be applied to model $fSCA$ at a daily temporal resolution. At a daily temporal resolution, the observed relationship between snow depth and $fSCA$ deviated from what Niu and Yang (2007) obtained for monthly $fSCA$. Mountainous terrain is not accounted for in the closed-form of Niu and Yang (2007). While the algorithm of Swenson and Lawrence (2012) empirically considers topography during ablation, their algorithm was, similar to that of Niu and Yang (2007), derived by linking satellite-retrieved $fSCA$ to snow data.

In contrast to Niu and Yang (2007); Swenson and Lawrence (2012) our $fSCA$ algorithm is developed for mountainous terrain using spatially measured snow depths at very high resolutions of a few meters. In order to describe realistic $fSCA$ following new snow and melt events throughout the season, we further track snow information with time at a high temporal resolution. We run the algorithm on hourly snow data, thus a much higher temporal resolution than for Niu and Yang (2007).

To perform a model intercomparison, we implemented the two closed-form parameterizations from Swenson and Lawrence (2012) as benchmark $fSCA$ model, as described in the technical description of the Community Land Surface model (CLM, version 5) (Lawrence et al., 2018). An evaluation of modelled $fSCA$ with our daily data sets showed that our $fSCA$ algorithm captures the seasonal evolution better than the CLM5.0 algorithm (cf. Table 3, 4, 5 and Figure 4, 5, 8 and 9).

It is true, that we apply some ad-hoc decisions for the seasonal algorithm, such as the 14 days time window for the detection of new snow amounts. We now provide more explanations for those decisions in Section 2 (description

of the $fSCA$ algorithm). We also mention our reasoning to apply the $\sigma_{HS}$ parameterization of Egli and Jonas (2009) for new snow events in mountainous terrain, though it was derived on snow depth values from spatially distributed flat field sites in mountainous terrain. While this approximation requires further investigation, coarse grid cells with a subgrid mean slope angle of zero are rare. For Switzerland we obtain a percentage of 0.01 %. Therefore, we could not reliably evaluate the performance of our algorithm for a flat grid cell. We suggest to use $\sigma_{HS}^{\textbf{Egli}}$ instead of $\sigma_{HS}^{\textbf{Helbig}}$ for a completely flat grid cell to avoid $fSCA = 1$ for a subgrid mean slope angle of zero (cf. Eq. (5)). However, for a global application of our algorithm any closed-form $fSCA$ parameterization could be applied for flat grid cells. We now mention that in the discussion.

There are indeed uncertainties involved when evaluating the seasonal $fSCA$ algorithm. For the evaluation, the algorithm was implemented in a comprehensive multilayer energy balance snow cover model which we ran with analysis data from an atmospheric model. This introduces model uncertainties to the algorithm performance (ranging from model input variables to uncertainties of other model equations). Additionally, the measurement data originate from various platforms adding observation uncertainties. Therefore, in order to focus on the performance evaluation of our $fSCA$ algorithm, we ideally have to minimize seasonal snow cover model or measurement uncertainties. As was already discussed in Section 5.2.3 removing grid cells with $HS < 5$ cm improved the performance statistics considerably. We now removed modelled $HS$ lower than 5 cm during pre-processing of the model data (Section 3.1). This reduced the scatter in Figure 8a (cf. new Figure 9a in the manuscript) and improved overall performance measures (Table 4,5).

Overall, we present an evaluation of a seasonal $fSCA$ implementation with independent high-resolution spatial as well as temporal snow depth data and snow products, something that has never been done for a seasonal $fSCA$ algorithm with such detail.

**References**

L. Egli and T. Jonas. Hysteretic dynamics of seasonal snow depth distribution in the Swiss Alps. *Geophys. Res. Lett.*, 36(L02501), 2009.

D. Lawrence, R. Fisher, C. Koven, K. Oleson, S. Swenson, and M. Vertenstein. Technical Description of version 5.0 of the Community Land Model (CLM), 2018. URL `https://www.cesm.ucar.edu/models/cesm2/land/CLM50_Tech_Note.pdf`. [Online; accessed 31-March-2018].

G. Y. Niu and Z. L. Yang. An observation-based formulation of snow cover fraction and its evaluation over large North American river basins. *J. Geophys. Res.*, 112(D21), 2007. doi: 10.1029/2007JD008674.

S. C. Swenson and D. M. Lawrence. A new fractional snow-covered area parameterization for the community land model and its effect on the surface energy balance. *Journal of Geophysical Research: Atmospheres*, 117(D21), 2012. doi: 10.1029/2012JD018178.

---

## Author Comment (AC2)

We thank the reviewer# 2 for the review and the comments. All comments (in italics) are addressed below (in bold).

**General comments**

*I found the evaluations of the algorithm generally convincing. The authors make use of a variety of evaluation data in order to assess the algorithm over a broad range of spatial and temporal scales and across a range of elevations. However the details of the algorithm implementation are not described clearly enough for the reader to follow. Furthermore, it's unclear whether the performance of the topographic algorithm is an improvement on existing algorithms that have been used to model $fSCA$ in mountain regions. Both of these issues should be addressed prior to publication.*

**Thank you very much for your comments and for pointing out the main two issues. We rewrote the description of the algorithm and also included two figures to more clearly explain how it works. Additionally, we now also include a comparison with the $fSCA$ algorithm of Swenson and Lawrence (2012) as benchmark $fSCA$ model, as described in the technical description of the Community Land Surface model (CLM, version 5) (Lawrence et al., 2018). Please see our detailed answers on both issues below.**

***mountainous vs flat terrain*** *Please be clear about whether you expect this algorithm to be applicable to nonmountainous regions (in the introduction and reiterate in discussions/conclusions) or whether it will be possible to merge it with other "flat" algorithms to make a global fSCA algorithm (I'm not sure to what extent equation 3 + equation 1 represents what is typically used in models over flat terrain). Do you anticipate that subgrid topographic parameters could be computed globally for every model grid cell and the parametrization used for flat grid cells as well? If so, does the expression for sigma_topo reduce to the "flat" sigma formula for perfectly flat terrain? If not, please state that you expect this algorithm as presently implemented is intended only to be used in simulations over mountainous terrain and it would require modification to implement it in a global climate model.*

**We completely rewrote the description of the $fSCA$ algorithm and also added two new figures for illustration and better understanding of our algorithm. As both the formulations for the standard deviation of snow depth were originally derived using data from mountainous areas, the labeling of the two with 'topo' and 'flat' in the original manuscript was somewhat misleading. We now changed the naming to the corresponding authors ('Helbig', 'Egli'), and more clearly describe the reasoning for using these two formulations.**
**Coarse grid cells with a subgrid mean slope angle of zero are rare. For Switzerland we only obtain a percentage of 0.01 %. Therefore, we could not re-**

liably evaluate the performance of our algorithm for a flat grid cell. We suggest to use $\sigma_{HS}^{\mathbf{Egli}}$ instead of $\sigma_{HS}^{\mathbf{Helbig}}$ for a completely flat grid cell to avoid $fSCA = 1$ for a subgrid mean slope angle of zero (cf. Eq. (5)). However, for a global application of our algorithm any closed-form $fSCA$ parameterization could be applied in our algorithm for a flat grid cell. We now mention that in the discussion.

*Your results comparing with ALS/ADS/camera/Sentinel data are well presented, but I would like to compare them with the typical performance of previous fSCA algorithms that have been applied in mountainous terrain. This may only require showing an extra row in your tables for how the "flat" parametrization (equation 3 only) performs relative to your combined eq 2+3, or a summary figure contrasting their performance. However, if there are other more standard parametrizations that have been used in mountain regions please consider demonstrating whether your algorithm is an improvement on those as well.*

To perform a model intercomparison, we implemented the two closed-form parameterizations from Swenson and Lawrence (2012) as benchmark $fSCA$ model as described in the technical description of the Community Land Surface model (CLM5.0) (Lawrence et al., 2018). An evaluation of modelled $fSCA$ on our daily data sets showed that our $fSCA$ algorithm captures the seasonal evolution better than the CLM5.0 algorithm (cf. Table 3, 4, 5 and Figure 4, 5, 8 and 9).

*algorithm description Section 2.4 (lines 110-123). This section only provides the reader with the most basic outline of how the algorithm works. I think this section needs to be broadened in particular with regards to how the seasonal and snow event aspects of the algorithm work together. A cartoon/schematic figure which illustrates several key decisions made within a two week window of time-varying HS and how those decisions affect the "seasonal" and "nsnow" curves would be extremely helpful. Ideally this schematic would highlight the differences between fSCA_season, fSCA_curr, and the full fSCA algorithm. At present I have no idea what the difference is between JIM_season and JIM_curr output because the point of the HS tracking has not been clearly communicated. For example, I don't understand how switching off new snow updates differs from performing no HS tracking? (line 143)*

We completely rewrote the description of the $fSCA$ algorithm and also added two new figures for illustration as suggested (new Figure 1 and 2). A new Table 1 gives an overview over the various $fSCA$ model simplifications in JIM$_{\mathbf{OSHD}}$ and the CLM5.0 benchmark $fSCA$ model as well as how they differ from our algorithm.

*Appendix The pseudocode and text currently provided in the appendix aren't clear enough to communicate the implementation of the algorithm. If you intend to publish the*

*complete algorithm (see the request to clarify this below), I would suggest you focus on providing a full description of the concepts/decision-making that it uses rather than pseudocode. The main text should communicate a basic understanding of how the algorithm operates (this is not the case presently), with further details deferred to the appendix, if you wish. While I think it would be helpful to provide a clearer version of the pseudocode along with the published algorithm (since the algorithm will presumably be provided in a specific computer language), I'm not sure it needs to be included in the paper if the rest of the description is sufficient. Please provide definitions for all terms. E.g. what does "recent" mean? What does "current" mean? Does recent snow = current snow? The treatment of melting (which I presume is tracked to remove the flat snow layer before reverting to the underlying topography-dependent layer) is unclear. The reason for tracking HS differences is never fully articulated. Again, I think that some sort of visual depiction/description of what's going on would be extremely valuable.*

**As outlined above, we completely rewrote the description of the $fSCA$ algorithm and also added two new figures for illustration (new Figure 1 and 2). For instance, Figure 1 shows all terms in context. With the new description of the algorithm and the new figures, we didn't see the need for pseudo-code anymore. Furthermore, we published the algorithm code on a gitlab repository and linked it to EnviDat, an environmental data portal.**

**Specific comments**

*Line 50: You introduce the idea of hysteresis here, but don't explicitly state that your algorithm includes it until Section 2. I think it's worth mentioning in the paper description at lines 62-73.*

**Thanks, we now mention this in the paper description.**

*Why was 2 weeks chosen as the period to track new snow and melted snow over? This may only cover 1 synoptic scale event – is that sufficient?*

**Testing of the algorithm showed that a two week window provided reliable simulations results of $fSCA$. However, this is still an ad-hoc decision which might indeed require further investigation once we know more about changes in snow depth distributions after snowfall. We now mention this in the algorithm description in the manuscript.**

*I realize you are using the "flat" parametrization to approximate a uniform blanketing of new snow, however the scatter at low elevations (fig 8) suggests that there may be better alternatives (although I'm not sure what level of agreement can be expected between modelled and observed $fSCA$ at such elevations – a comment on this would be useful).*

[Figure]

Figure 1: Sentinel-derived $fSCA$ minus modelled $fSCA$ without any $fSCA_{nsnow}$ (JIM$_{\text{OSHD}}^{\text{season}}$) for Switzerland as a function of date and elevation $z$ for available satellite dates.

**Most of the scatter in Figure 9 (formerly Fig. 8) is actually tied to low $HS$ at lower elevations or along the snow line. This may point to problems with the $fSCA_{\text{nsnow}}$ in our algorithm. However, the scatter remains when we neglect $fSCA_{\text{nsnow}}$ and only compute $fSCA_{\text{season}}$ using $\sigma_{HS}^{\text{Helbig}}$ via Eq. (2)-(3) (Figure 1), suggesting that $fSCA_{\text{nsnow}}$ is not the main reason for this scatter.**

**We therefore assume that most of the $fSCA$ scatter at low elevations (Figure 9) originates from modelled $HS$ errors when snow falls on bare ground, i.e. early in the season or at the snow line. As discussed in Section 5.2.3, removing grid cells with $HS < 5$ cm improved the performance statistics considerably. In order to focus on the performance evaluation of our $fSCA$ algorithm, ideally we should minimize seasonal snow cover model or measurement uncertainties. Therefore, we now also removed modelled $HS$ lower than 5 cm during pre-processing of the model data (Section 3.1). This reduced the scatter in Figure 8a (cf. new Figure 9a in the manuscript) and improved overall performance measures (Table 4,5).**

*Please confirm: HS=HS(X,t), where X is the location on the coarse model grid (with grid size, L from eq 2) and t is the time (day of the year, for example). Likewise in equations 2 and 3 the HS variables based on the temporally and spatially varying values of HS from the model (hence one could substitute eqs 2 and 3 into eq 1 and simplify to get two forms of eq 1) – is that correct?*

**Your description is correct. We now present the three $fSCA$ ($fSCA_{\text{season}}$, $fSCA_{\text{nsnow}}^{\text{recent}}$ and $fSCA_{\text{nsnow}}^{\text{14day}}$) as suggested (cf. Eq. (5)-(7)).**

*Fig 1: By "reset" fSCA_season do you mean that you assign the fSCA_nsnow value to fSCA_season?*

**The description was unclear. We meant to say that $fSCA_{\mathrm{nsnow}}$ is set to zero and $fSCA_{\mathrm{season}}$ is recalculated using a newly estimated seasonal $HS_{\mathrm{max}}$. However, this reset function was removed in order to simplify the algorithm. All model results were recalculated accordingly.**

*Figure 3+4: the colors of the "red and blue stars" appear close to orange and purple to me – especially if the page is zoomed out. Can you adjust the colors or the text to be slightly more in line with each other? Adjusting the red stars closer to orange may actually be more helpful since it would further distinguish it from the red circles of the model output data.*

**Changed as suggested.**

*Line 327: Your argument isn't 100% clear here – isn't it an easy comparison to provide results using only sigma_topo and to explicitly see if the flat parametrization of new snow events is helpful?*

**Thank you for the suggestion. We made an additional model run for the winter season 2017/18 to evaluate model performance with Sentinel-derived $fSCA$ only using $\sigma_{HS}^{\mathbf{Helbig}}$ to derive $fSCA$ (JIM$_{\mathbf{OSHD}}^{\mathbf{allHelbig}}$). Model performance was similar. Thus, while applying $\sigma_{dHS}^{\mathbf{Egli}}$ might not describe the true spatial new snow distribution in mountainous terrain, it is a first approach. Furthermore, $\sigma_{dHS}^{\mathbf{Egli}}$ is required for flat grid cells, where $\sigma_{HS}^{\mathbf{Helbig}}$ is always 1. We now mention these points at the beginning of the discussion (Section 5.1).**

*Line 350: Rephrase. It's unclear what you mean by "modelled fSCA does not show similar strong trends when compared to Sentinel-derived fSCA..."*

**Thanks, we rephrased that.**

*Line 386: do you mean "versus Table 3, I"?*

**Yes, we meant Table 3, I. Thanks.**

*Line 387-407: Your points about deriving fSCA from camera images may be valid, however, the ALS and ADS evaluations also represent a spatially averaged evaluation at a single time generally closer to mid-season, while the camera evaluation represents a near-point location evaluated continuously including the very beginning and end of the season. Hence the difference in NRMSE could also represent a true difference in the ability of the algorithm to*

*capture snow cover on average versus at a single location. Or how performance varies with HS.*

Though ALS/ADS-derived $fSCA$ was only available for **6 points in time**, at least one data acquisition date was during late ablation (**17 May**) and one towards earlier accumulation (**26 January**). We selected camera images to obtain $fSCA$ evaluation data in a much higher temporal resolution than the **6 points in time** of the ADS/ALS data sets. Camera images allowed us performing a much more continuous temporal evaluation of modelled $fSCA$- though for three grid cells only.

We agree that the larger temporal resolution of camera-derived $fSCA$ reveals larger $fSCA$ performance issues due to the spatially and temporally varying performances of modelled $HS$. We already mentioned this in the discussion of the evaluation with Sentinel-derived $fSCA$ (Section 5.2.3) where we obtained performance improvements by **10 %** when neglecting grid cells with $HS$ lower than **5 cm**. We now also removed modelled $HS$ lower than **5 cm** during pre-processing of the model data (Section 3.1). While this improved the overall scatter for the evaluation with Sentinel-2 data considerably, performances with camera-derived $fSCA$ improved only slightly. We therefore agree that a difference in the NRMSE could indeed also origin from the fact that for the evaluation with camera-derived $fSCA$ we only have three grid cells instead of several hundreds to thousands. This makes it clearly more susceptible to outliers.

A large difference between ALS, ADS-derived $fSCA$ and camera-derived $fSCA$ are however also the uncertainties related to the product itself. Besides acquisition uncertainties, it matters if a grid cell representative $fSCA$ value is derived allowing for **30 % NaN** or **90 %**. Additionally, a camera "sees" more the steep mountain faces from a slope as opposed to which the flatter parts remain invisible. Unfavorable weather conditions during the longer covered time periods of camera increased product uncertainties.

We extended the discussion on this in Section 5.2.2.

*Code: availability: Does the "depletion curve implementation" differ from the full algorithm?*

No it doesn't. We rephrased that.

**Technical comments**

*The paper contains a lot of non-standard English grammar to the point that it affects reading comprehension. The paper would benefit from a professional copy-editing service.*

Thanks, we carefully went through the manuscript and improved the lan-

**guage.**

**References**

D. Lawrence, R. Fisher, C. Koven, K. Oleson, S. Swenson, and M. Vertenstein. Technical Description of version 5.0 of the Community Land Model (CLM), 2018. URL `https://www.cesm.ucar.edu/models/cesm2/land/CLM50_Tech_Note.pdf`. [Online; accessed 31-March-2018].

S. C. Swenson and D. M. Lawrence. A new fractional snow-covered area parameterization for the community land model and its effect on the surface energy balance. *Journal of Geophysical Research: Atmospheres*, 117(D21), 2012. doi: 10.1029/2012JD018178.

---

## Author Response (AR2)

We thank the reviewer# 1 for the review and the comments. All comments (in italics) are addressed below (in bold).

*This revision is greatly improved. I have some minor points:*
**Thank you very much for your second review.**

*10 It is not clear what these errors respectively refer to.*
**We rewrote the description of the data sets and errors.**

*38 Were any of these fSCA parameterizations \*not\* heuristically developed? What is the intended point here?*
**We rephrased this to focus on the heuristic $tanh$-form for $fSCA$ from Yang et al. (1997) a form which was later theoretically confirmed by Essery and Pomeroy (2004). In fact, the parameterizations of Swenson and Lawrence (2012) and Niu and Yang (2007) were empirically based.**

*43 "allowing empirical parameterization of"*
**Thanks. Changed.**

*129 Reference to Eq. (3) should be (2).*
**Corrected. Thanks.**

*166 "computationally efficient"*
**Corrected. Thanks.**

*Table 2 Spatial coverage for the camera is incorrect.*
**The spatial coverage for the camera data set of 931 km$^2$ is correct and was obtained by summing all valid 1 km$^2$ grid cells over the 5 seasons. This is described in Section 3.3 and is referred to in the caption of Table 2.**

*321 "after snowfall events"*
**Corrected.**

*327 worst overall performances"*
**Corrected.**

*Figure 8 The Swenson model results have been added in revision and listed at the end of the figure legend; this would make more sense above "camera" and "Sentinel". Putting the figure letters in the lower left would avoid overlap with data points in (c). I don't understand why the last sentence has been added to the figure caption.*
**We changed the order of the legend entries (as well as for Fig. 4 and 5) and put the figure letters in the lower left corner. The last sentence of the figure**

**caption was removed. It appeared due to a copy error. We apologize for this.**

*Could much better performance of the Swenson algorithm be obtained by simply changing the Nmelt parameter? (I haven't checked, but max would seem to make more sense than min in the denominator of Lawrence et al. equation 8.3)*
**We evaluated the Community Land Model (CLM5.0) $fSCA$ algorithm as it is described in Lawrence et al. (2018) and which is based on the Swenson and Lawrence (2012) algorithm. We have not checked if modifications of the presented algorithm implementation would improve its performance.**

*418 "likely also due to"*
**Corrected.**

*457 "poorer than those"*
**Corrected.**

**References**

R. Essery and J. Pomeroy. Implications of spatial distributions of snow mass and melt rate for snow-cover depletion: theoretical considerations. *Ann. Glaciol.*, 38, 2004.

D. Lawrence, R. Fisher, C. Koven, K. Oleson, S. Swenson, and M. Vertenstein. Technical Description of version 5.0 of the Community Land Model (CLM), 2018. URL `https://www.cesm.ucar.edu/models/cesm2/land/CLM50_Tech_Note.pdf`. [Online; accessed 31-March-2018].

G. Y. Niu and Z. L. Yang. An observation-based formulation of snow cover fraction and its evaluation over large North American river basins. *J. Geophys. Res.*, 112(D21), 2007. doi: 10.1029/2007JD008674.

S. C. Swenson and D. M. Lawrence. A new fractional snow-covered area parameterization for the community land model and its effect on the surface energy balance. *Journal of Geophysical Research: Atmospheres*, 117(D21), 2012. doi: 10.1029/2012JD018178.

Z. L. Yang, R. E. Dickinson, A. Robock, and K. Y. Vinnikov. On validation of the snow sub-model of the biosphere atmosphere transfer scheme with russian snow cover and meteorological observational data. *J. Climate*, 10(2):353–373, 1997.